# Managing fire risk during drought: the influence of certification and El Niño on fire-driven forest conversion for oil palm in Southeast Asia

Praveen Noojipady[1, 2], Douglas C. Morton[1], Wilfrid Schroeder[2], Kimberly M. Carlson[3], Chengquan Huang[2], Holly K. Gibbs[4], David Burns[5], Nathalie F. Walker[5], Stephen D. Prince[2]

[1]NASA Goddard Space Flight Center, Greenbelt, MD 20771, USA
[2]University of Maryland, College Park, MD 20742, USA
[3]University of Hawai'i, Honolulu, HI 96822, USA
[4]University of Wisconsin, Madison, WI 53706, USA
[5]National Wildlife Federation, National Advocacy Center, Washington, DC 20006, USA

*Correspondence to*: Praveen Noojipady (praveen.noojipady@nasa.gov)

**Abstract.** Indonesia and Malaysia have emerged as leading producers of palm oil in the past several decades, expanding production through the conversion of tropical forests to industrial plantations. Efforts to produce "sustainable" palm oil, including certification by the Roundtable on Sustainable Palm Oil (RSPO), include guidelines designed to reduce the environmental impact of palm oil production. Fire-driven deforestation is prohibited by law in both countries and a stipulation of RSPO certification, yet the degree of environmental compliance is unclear, especially during El Niño events when drought conditions increase fire risk. Here, we used time series of satellite data to estimate the spatial and temporal patterns of fire-driven deforestation in and around oil palm plantations. In Indonesia, fire-driven deforestation accounted for one quarter of total forest losses in both certified and non-certified plantations. After the first plantations in Indonesia received RSPO certification in 2009, forest loss and fire-driven deforestation declined in certified plantations but did not stop altogether. Oil palm expansion in Malaysia rarely involved fire; only 5% of forest loss in certified plantations had coincident active fire detections. Interannual variability in fire detections was strongly influenced by El Niño and the timing of certification. Fire activity during the 2002, 2004, and 2006 El Niño event was similar among oil palm plantations in Indonesia that would later become certified, non-certified plantations, and surrounding areas. However, total fire activity was 75% and 66% lower in certified plantations than non-certified plantations during the 2009 and 2015 El Niño events, respectively. The decline in fire activity on certified plantations, including during drought periods, highlights the potential for RSPO certification to safeguard carbon stocks in peatlands and remaining forests in accordance with legislation banning fires. However, aligning certification standards with satellite monitoring capabilities will be critical to realize sustainable palm oil production and meet industry commitments to zero deforestation.

## 1 Introduction

Global production of agricultural commodities such as palm oil has risen steadily in recent decades in response to market demand (USDA, 2009 , 2010 , 2016). Southeast Asia's palm oil sector has grown through expansion of oil palm plantations in Malaysia, Indonesia, and more recently, Papua New Guinea (Gunarso et al., 2013; Carlson et al., 2013; Miettinen et al., 2016a; Vijay et al., 2016). By 2014, Indonesia and Malaysia accounted for nearly 69% of harvested oil palm area worldwide (FAO, 2016).

In the past decade, Indonesia had the highest rate of forest loss of any country in Southeast Asia (Hansen et al., 2013; Margono et al., 2014; Kim et al., 2015), spurred by rapid forest conversion for oil palm and other industrial plantations (Carlson et al., 2012; Gunarso et al., 2013; Abood et al., 2015).  Between 1990-2010, more than one third of new oil palm plantations replaced forested landscapes in Southeast Asia (Gunarso et al., 2013; Gaveau et al., 2016), with rates as high as 90% in regional hotspots such as coastal West Kalimantan, Indonesia (Carlson et al., 2013). Conversion of primary and logged forests for oil palm, including vast areas with deep organic peat soils, contributed to significant greenhouse gas (GHG) emissions from fire, decomposition, and peat oxidation (Page et al., 2002;  van der Werf et al., 2008; Hooijer et al., 2012; Ramdani and Hino, 2013; Field et al., 2016; Huijnen et al., 2016). Environmental concerns with palm oil production extend beyond GHG emissions, however, as forest loss threatens biodiversity (Pimm et al., 2014; Vijay et al., 2016) and particulate emissions from fires are a major public health concern in Indonesia and downwind population centers such as Singapore (Murdiyarso et al., 2004; Gaveau et al., 2014; Kunii et al., 2002; Reddington et al., 2014; Marlier et al., 2015; Chisholm et al., 2016; Johnston et al., 2015).

Palm oil is the fastest growing certified agriculture commodity, and Indonesia accounted for >50% of certified production areas in 2016 (Potts et al., 2014; RSPO, 2016). The push for certification within the palm oil industry reflects a growing consumer awareness of GHG emissions from palm oil expansion and peat oxidation and an overall rise in producer and consumer interest in "sustainable" and deforestation-free products (UNCS, 2014; Butler, 2015; McCarthy et al., 2016). The Roundtable on Sustainable Palm Oil (RSPO) certification is the most widely adopted certification standard.  By 2016, the RSPO had certified 2.83 Mha of oil palm that produced 10.8 million tons of palm oil, or approximately 17% of global palm oil production, with >90% of certified areas in Southeast Asia (RSPO, 2016).  Specific principles and criteria of RSPO certification promote sustainable palm oil production and processing (Garrett et al., 2016; RSPO, 2004, 2015b). Among other provisions, RSPO certification prohibits conversion of primary and high conservation value (HCV) forests and bans fire use for land clearing in compliance with the Indonesian moratorium on fire (RSPO, 2007; RSPO, 2013 ; Edwards and Heiduk, 2015).  Companies interested in certification first become members of the RSPO and are subjected to the Principles & Criteria (P&C), including the prohibition on new plantings through deforestation of primary or HCV forests after Nov. 2005 without compensation (Criterion 7.3, RSPO 2013). However, the RSPO does not independently monitor deforestation within member plantations.  RSPO members companies are notified of satellite fire detections on their plantations, and a description of fire incidents is included in monthly reports to RSPO. Recent fire activity has been assessed for a subset of

palm oil concessions in Indonesia (Cattau et al., 2016), but the use of fire for forest conversion on oil palm plantations has not previously been quantified.

Improving estimates of fire-driven deforestation is critical to assess environmental compliance by oil palm plantations, reduce uncertainties in deforestation carbon emissions (Le Quéré et al., 2015; Houghton et al., 2012; van der
Werf et al., 2009b), and characterize ignition sources that may give rise to uncontrolled burning during drought periods (Carlson et al., 2012; Cattau et al., 2016). The timing of GHG emissions from forest conversion to oil palm depends on the degree of fire use for deforestation (DeFries et al., 2008; Houghton et al., 2012), including the proportion of clearing activity through fire and the combustion completeness of initial or repeated burning (van der Werf et al., 2009a). Fires are common in industrial plantations and smallholder properties (Stolle et al., 2003; Austin et al., 2015; Marlier et al., 2015; Miettinen et
al., 2016b; Cattau et al., 2016), yet the link between fire activity and forest conversion is unclear. Many estimates of carbon emissions from tropical forest conversion report committed fluxes without separating fire and decomposition losses (Koh et al., 2011; Carlson et al., 2012; Austin et al., 2015). Previous studies with bookkeeping or biogeochemical models suggest that fire accounts for 30% (Houghton and Hackler, 1999) to 50% (van der Werf et al., 2009a) of carbon emissions from aboveground biomass during forest conversion in Southeast Asia—a broad range that applies to all forest conversion, not
strictly to oil palm expansion. Fires are not restricted to forested areas; El Niño conditions suppress precipitation over large parts of Southeast Asia, leading to widespread fire activity during drought periods, particularly in carbon-rich peatlands (Page et al., 2002; van der Werf et al., 2008; Field et al., 2009, 2016). Understanding the contribution from fire-driven deforestation to total fire activity is therefore a critical part of mitigating fire risk during drought years (e.g., Chen et al., 2016).

Here, we combined time series of satellite data on forest loss and active fire detections with locations of oil palm plantations to assess fire-driven forest and peatland conversion in and around oil palm plantations. The combination of land management, forest loss, and active fire data provided an opportunity to evaluate the relative contributions from different fire types to spatial and temporal variability in satellite fire detections. Our study addressed three primary questions regarding oil palm expansion: 1) What fraction of forest and peat forest conversion for oil palm involves fire? 2) Does certification
alter fire use for forest conversion or the frequency of management or accidental fires in plantation areas? and 3) During El Niño years, do certified plantations have fewer satellite fire detections compared to non-certified plantations and surrounding lands? Characterizing fire-driven deforestation is critical to evaluate the influence of RSPO certification on fire activity and to improve estimates of GHG emissions from oil palm expansion.

## 2 Material and Methods

### 30 2.1 Oil Palm Plantations

The government of Indonesia allocates land for oil palm production to companies for a limited period of time. We separated leases for oil palm production into two categories, certified and non-certified plantations. Certified plantations are properties

certified by the RSPO; non-certified plantations are properties allocated by the Indonesian government to companies but that are not certified, even if they are held by RSPO members. Comparisons between certified (ever) and non-certified (never) plantations considered forest loss and fire activity over three time scales: 1) following the benchmark date for compliance with RSPO Criterion 7.3 (Nov. 2005), 2) following the first issuance of RSPO certificates to Indonesian producers in 2009,
and 3) following the date of certification for individual plantations. Boundaries of certified plantations were compiled from several sources, including boundary polygons provided by the RSPO, digitized boundaries from RSPO audit reports, and spatial data on plantation boundaries from RSPO member companies provided in annual communication of progress (ACOP) reports (RSPO, 2015a). Boundaries of non-certified plantations were obtained from a database of oil palm plantations published by Greenpeace (Greenpeace, 2016) and supplemented with non-certified plantations held by RSPO
members, as indicated in ACOP reports (RSPO, 2015a) or by Sawit Watch (2013). In total, we analysed 154 certified and 1536 non-certified plantations boundaries for Indonesia (Fig. 1). Data on the location and certification date of certified plantations were also available for Malaysia (n =119) and Papua New Guinea (n = 10), but boundaries of non-certified plantations were not available.

We used maps of planted oil palm to identify established plantations within certified and non-certified plantations in
Indonesia, Malaysia, and Papua New Guinea. Data on the extent of planted oil palm were compiled from three sources: Gunarso et al. (2013) for 2000, 2005, and 2010; Carlson et al. (2013) for 2000, 2005, and 2010; and Transparent World (TW, 2015) for 2014. Maps of planted palm were generated from 30 m Landsat imagery, and validated using higher-resolution satellite imagery (Carlson et al., 2013; TW, 2015; Petersen et al., 2016). When multiple estimates were available for the same epoch, we used the combined area from all sources as a more conservative estimate of the extent of planted oil palm.
Only non-certified plantations with evidence of planted oil palm by 2014 were included in this study.

We evaluated forest loss and fire activity for a single set of 5 km buffers surrounding both certified and non-certified plantations in Indonesia. Recent fire emergencies have intensified the debate over the source of fire ignitions during El Niño and other drought events (e.g., Austin et al., 2015). The 5km buffer was chosen to capture the potential influence of forest loss and fire activity on the surrounding landscape, including the direct fire spread from adjacent lands
into palm oil plantations, wind-blown embers from nearby fires, and the most acute impacts of smoke on both human health and ecosystems. Oil palm plantations in Southeast Asia are frequently adjacent to other oil palm plantations (Fig. 1), making it difficult to attribute buffer activities to only certified or non-certified neighbours. We therefore analysed a single set of buffer areas to evaluate forest loss and fire activity surrounding large oil palm plantations. Nearly 12% of the area within the 5km buffer was mapped as planted oil palm in 2010 (Gunarso et al., 2013; Carlson et al., 2013). As for plantations, areas of
planted oil palm were excluded from estimates of forest loss in buffer areas. However, total fire activity in the buffer region may reflect differences in oil palm management, in addition to differences in land use and land cover, based on the abundance of planted palm outside of the large certified and non-certified plantations.

**2.2 Forest definition, cover, and loss**

Estimates of forested area and forest loss fundamentally depend on the definition of forest cover (Sexton et al., 2016). Countries may use canopy cover thresholds between 10-30% for reporting under the United Nations Framework Convention on Climate Change (UNFCCC) REDD+ framework (UNFCCC, 2002). The Indonesian government uses two definitions of

forest for reporting purposes (BP-REDD+, 2015). For the United Nations Food and Agriculture Organization (FAO) Forest Resource Assessment (FRA), forest is defined as canopy cover $\geq$ 10% for Global Forest Resource Assessment (FAO, 2010), whereas the Minister of Forestry Decree uses a higher threshold of canopy cover > 30% for Forest Reference Level Emissions (FREL) reporting (MoF, 2008), at the high end of the REDD+ range. Therefore, we used the >30% canopy cover threshold in this study to be consistent with the forest definition used by the Indonesian government. The higher canopy

cover threshold also reduced ambiguity associated with discriminating tropical forests from other land cover types in remote sensing data for regions with persistent cloud cover, such as Southeast Asia. Forest and non-forest areas were separated using Landsat-based estimates of fractional tree cover in 2000 (Hansen et al., 2013). Estimates of annual forest loss between 2002-2014 (Hansen et al., 2013) were used to identify the timing of forest conversion in and around plantations.

**2.3 Active fires**

We used the time series of active fire detections from the Moderate Resolution Imaging Spectroradiometer (MODIS) instruments on NASA's Terra and Aqua satellites to evaluate the spatial and temporal patterns of daily fire activity during 2002-2015. The global monthly fire location product (MCD14ML) identifies the location of actively burning fires and thermal anomalies at the time of satellite overpass at 1 km nominal spatial resolution (Giglio et al., 2003). Fire pixel counts from Terra and Aqua MODIS sensors were combined using a 1km grid to evaluate monthly and annual fire activity from

2002 to 2015. We compared fire pixel density ($km^{-2}$) across certified plantations, non-certified plantations, and a 5km buffer region surrounding both certified and non-certified plantations.

For 2014 and 2015, higher spatial resolution active fire detections were used to confirm patterns in 1 km MODIS fire data. These complementary active fire detections were derived from the Visible Infrared Imaging Radiometer Suite (VIIRS) I-band (375m) on the Suomi-National Polar orbiting Partnership (S-NPP) (Schroeder et al., 2014) and Landsat-8

Operational Land Imager (OLI) data at 30 m resolution (Schroeder et al., 2015). The finer spatial resolution of these fire data capture additional details regarding fire activity that can be difficult to evaluate at MODIS resolution, including the precise location of active fire fronts, separation of flaming and smouldering fires (Elvidge et al., 2015), and detection of small and/or lower intensity fires (Schroeder et al., 2015)—an important component of fire activity in agricultural landscapes (Randerson et al., 2012). In this study, the improved spatial resolution of VIIRS and OLI fire data aided the attribution of

active fires to specific land management areas.

Praveen Noojipady 7/18/2017 11:48 AM
**Deleted:** $\geq$
Praveen Noojipady 7/18/2017 11:57 AM
**Deleted:** 30

**1.4 Fire-driven forest conversion for oil palm expansion**

We combined satellite remote sensing data on forest cover (2000; Hansen et al., 2013), forest cover change (2002-2014; Hansen et al., 2013), and active fire detections (2001-2014; Giglio et al., 2003) to identify fire-driven forest conversion in certified and non-certified plantations. Our assessment excluded forested areas identified as oil palm (Gunarso et al. (2013);
Carlson et al. (2013)). Deforestation within oil palm plantations was therefore limited to Hansen et al. (2013) tree cover loss in forested areas (tree cover >30%) outside of planted palm. Oil palm expansion into peat forests was assessed using peatland layers created by Wahyunto et al. (2003, 2004, 2006) and Wetlands International (WI, 2016) (Fig. A1). Co-located forest loss and active fire detections were considered fire-driven deforestation. Given the potential for fire activity to pre-date the detection of forest loss (Morton et al., 2008), active fire data from the year of forest loss and one year before were
combined to identify fire activity associated with forest conversion.

**3 Results**

**3.1 Certification and Fire-driven Deforestation**

In Indonesia, forest loss in and around oil palm plantations reduced remaining forest cover by 30-36% between 2002-2014 (Fig. 2). Gross forest loss within plantations but outside of planted palm areas totalled 3.59 Mha (Table 1). Average annual
rates of forest loss were similar in certified (1.25% $yr^{-1}$) and non-certified plantations (1.72% $yr^{-1}$) over this period. However, trends in annual forest loss differed between certified plantations and both non-certified and buffer areas. Following the cut-off date for new deforestation (Nov. 2005), rates of forest clearing actually increased in plantations that would later be certified (2006-2008, 38,636 ha $yr^{-1}$) before declining sharply after the first certificates were issued (2009-2015, 10,943 ha yr-1, Table B2). Between 2009 and 2012, the majority of forest loss occurred on plantations that had not
yet received RSPO certification (Table B2), whereas plantations with certificates accounted for most forest losses identified in 2013-2014. Patterns of peat forest loss on certified plantations were similar to lowland forest loss outside of peat (Fig. 2, Table B3), with a peak in peat forest loss around the 2006 El Niño event followed by a steady decline after 2009. In contrast, rates of forest and peat forest loss in non-certified plantations increased over time, with peak clearing in 2009 and 2012 (Figure 2, Tables B2, B3). Temporal patterns of forest loss for buffer areas within 5 km of plantations (both certified
and non-certified) were similar to non-certified plantations. Given the larger extent of non-certified plantations, mean annual forest losses differed by more than order of magnitude between certified and non-certified plantations (20,610 ha $yr^{-1}$ and 194,070 ha $yr^{-1}$, respectively).

      Although the use of fire for forest conversion is prohibited in Indonesia, satellite data suggest that one quarter of forest clearing in both certified and non-certified plantations involved fire (Table 1). For certified plantations in Indonesia,
the proportion of fire-driven deforestation in both lowland and peat forests declined sharply following the cut-off date for new deforestation (Nov. 2005) but before the first RSPO certificates were issued (Fig. 2), from 37% during 2002-2005 to

19% in 2006-2008, and only 8% of all forest loss was identified as fire-driven deforestation after the first certificates were issued (2009-2014, Tables B2, B3). More than 1/3 of Indonesian plantations that would later be certified had fire-driven deforestation between 2002-2007 (Table B4). As total deforestation declined from 2008 to 2014, fire-driven deforestation activity was still distributed across 34-50 plantations (22-32%, Table B4). The proportion of fire-driven deforestation in

non-certified plantations also declined over time, but not as rapidly as in certified plantations, especially in peat areas. Fire-driven forest losses accounted for 14% of total forest loss in non-certified and buffer areas from 2009-2014 (Table B2), but 34% of peat forest loss was identified as fire-driven deforestation (Table B3). Notably, the proportion of fire-driven deforestation in El Niño years (2002, 2006, 2009) was similar to or lower than non-El Niño years (e.g., 2003, 2007, 2010) for all three management types—certified plantations, non-certified plantations, and buffer areas.

However, certification did not halt forest conversion altogether. In Indonesia, forest loss continued within certified plantations following the start of RSPO certification efforts, including fires for forest conversion, leading to an additional 8% loss of remaining forest cover between 2009-2014 (Fig. 2, Tables B2, B3, B4). Lower rates of forest loss on certified plantations are consistent with RSPO restrictions on clearing HCV forest areas and other lands deemed unsuitable for palm oil production. Declining rates of forest loss after 2009 may also reflect limited remaining forest cover on certified

plantations by 2014 (15%; Fig. 2), leading to smaller clearing sizes that are more difficult to assess with remote sensing data on forest loss and fire activity (Fig. 3). In contrast, the contribution from larger clearing sizes increased over time in non-certified plantations and remained stable for buffer areas.

        Patterns of fire-driven forest loss in certified plantations differed across Indonesia, Malaysia, and Papua New Guinea (Fig. 4). Overall forest loss rates were higher in Indonesia than Malaysia and Papua New Guinea (Table B1).

However, large forest clearing events were more common on certified plantations in Malaysia and Papua New Guinea, with more than two-thirds of forest loss in patches >10 ha (Fig. A2). Annual forest loss rates in Malaysia remained high following certification, with little change from pre-certification patterns (Fig. 4, Tables B5, B6). In Malaysia, oil palm expansion in certified plantations rarely involved fire, and only 5% of total forest loss was identified as fire-driven deforestation (Tables B4, B5, B6). Fire detections associated with forest loss declined in all three countries following the start of certification in

2008-2009. In Malaysia and Papua New Guinea, plantations with RSPO certificates had little fire-driven deforestation and few total fire detections for land management (Tables B4 – B7).

        Certification decoupled fire detections from ENSO-driven variability in fire risk in Indonesia. Interannual variability in Indonesian fire activity is largely governed by the timing and magnitude of El Niño events (Fig. A3; Chen et al., 2016). Prior to 2009, interannual variability in fire detections was similar for certified plantations, non-certified

plantations, and buffer areas in Indonesia (Fig. 5). Mean fire rates across land management classes were also consistent during El Niño events in 2002, 2004, and 2006, with important contributions from fire-driven deforestation to total fire detections in these years. Following certification, fire activity declined in certified plantations in all years, with 75% and 66% fewer fires km$^{-2}$ than non-certified plantations during the 2009 and 2015 El Niño events, respectively (Figure 5, Table B6). Monthly fire counts confirmed the reduction in fire activity within certified plantations during peak burning months

(August, September, October) of the 2009 and 2015 El Niño events (Fig. A4). Evidence for reduced fire activity in certified plantations highlights the potential for management of fire risk within oil palm plantations, even during strong El Niño drought conditions (Fig. A5).

        Attribution of fire activity is a critical component of satellite-based monitoring for environmental compliance.
Higher resolution active fire data from VIIRS (375 m) and Landsat 8/OLI (30 m) confirmed the relative decline in fire activity on certified plantations compared to non-certified plantations and buffer areas in both 2014 and 2015 (Fig. 6). The VIIRS 375 m fire data provided a more complete characterization of the fire perimeter than MODIS on a daily basis. Although less frequent, Landsat 8 coverage every 16 days captured the precise location of active fire fronts, small fires, and persistent smouldering in peat areas that may last for many days (Fig. 6 and Fig. A6). High-resolution fire data offer
improved understanding of fire use for deforestation and agricultural management, with detections that can be more definitively attributed to specific actors in support of monitoring, reporting, and verification.

## 4 Discussion

Following the issuance of the first RSPO certificates in 2009, certified oil palm plantations in Indonesia had lower fire-driven deforestation and total fire activity than non-certified plantations, including during El Niño events.  These reductions
point to the potential for RSPO to contribute to REDD+ and to decrease fire ignitions during drought conditions.  Our findings of lower fire activity on certified plantations for both deforestation and land management during the 2009 and 2015 El Niño events contradicts earlier work by Cattau et al. (2015) showing higher fire activity on a small subset certified plantations, possibly due to a larger sample size of certified plantations in our study (N=154 compared to 28, with only 4 plantations on peat).  However, certification did not halt forest losses or fire activity altogether, and rates of forest loss
actually increased following the cut-off date for new deforestation but before the first RSPO certificates were issued. In addition, certified plantations currently account for a small fraction of total oil palm leases (e.g., 13% in Indonesia); non-certified plantations maintained higher rates of fire-driven deforestation and fire activity in recent years, including the 2015 El Niño.  The opportunity exists, therefore, to enhance the environmental benefits of RSPO certification through expansion of certified plantations and strengthening of certification standards, including the use of satellite monitoring of fire activity
and forest loss.

        Our study confirmed the pervasive use of fire for forest conversion to oil palm in Indonesia, with one quarter of forest loss identified as fire-driven deforestation.  Fire-driven deforestation was less common on certified plantations in Malaysia and Papua New Guinea, and fire use for forest conversion declined to near zero after the start of certification in 2008-2009 in these countries.  The long-term records of Landsat forest loss and MODIS fire detections provide robust
evidence of changing fire use for land management in certified oil palm plantations.

        Several factors may account for the reduction in fire activity on certified plantations following certification.  First, certification may reduce fire-driven deforestation by directly influencing land management practices.  Collectively, all

certified plantations in Indonesia, Malaysia, and Papua New Guinea showed declines in fire-driven forest losses after 2009. Second, the observed decline in fire activity may indicate an end of the expansion process rather than a change in fire-driven deforestation. Remaining forest cover was only 8-15% on certified plantations in Malaysia and Indonesia; remaining forest areas may not be suitable for oil palm or accessible based on RSPO restrictions. In a future study, it may be possible to control for differences in remaining forest cover, plantation age, or company management practices using a matched study design. For Indonesia, a reduction in overall fire activity may be less important for GHG emissions than a reduction in peat fires (e.g., van der Werf et al., 2008; Cattau et al., 2016; Field et al., 2016). Regardless, the potential exists for RSPO to promote fire-free management of plantations to protect high-value tree crops and remaining carbon stocks in forests and peatlands. Large labor forces needed for oil palm production (Lambin et al., 2013) may aid regional fire suppression efforts, allowing established plantations to maintain lower fire activity in and around plantations during El Niño years.

The proportion of fire-driven deforestation on oil palm plantations in Indonesia (~25%) was similar to the estimate of combustion losses in bookkeeping models (30-40%; Houghton and Hackler, 1999), but fire use was much lower in Malaysia and Papua New Guinea. However, our study only confirms the coincident timing and locations of fires and forest losses, not the combustion completeness of fires for forest conversion. Removal of forest vegetation is critical to establish an oil palm plantation, but combustion completeness may be lower for these fires, given higher fuel moisture and less need for complete combustion of aboveground biomass than for expansion of row crop agriculture (Morton et al., 2008). Fuel moisture also has a substantial influence on trace gas emissions from fire, including smouldering fires in peatlands (Miettinen et al., 2012; Page and Hooijer, 2016). By combining active fire detections with satellite observations of trace gas emissions, it may be possible to characterize regional GHG emissions directly associated with fires on oil palm plantations. Complementary data on trace gas emissions may also compensate for missing satellite fire detections. The estimated fraction of fire-driven deforestation for different land management categories in this study is likely conservative because satellite platforms do not detect all fires. Satellite sensors may not sample at the peak of diurnal fire activity (Giglio et al., 2000), and cloud obscuration (Giglio et al., 2003) and orbital coverage (Schroeder et al., 2005) reduce the probability of fire detections, particularly for low-latitude regions with persistent cloud cover such as Southeast Asia. New satellite products partially overcome these limitations through improvements in orbital coverage and spatial resolution (Schroeder et al., 2014), especially for detection of small and low-intensity fires in deforestation or peatland areas (Schroeder et al., 2015; Elvidge et al., 2015).

Aligning certification criteria with existing satellite monitoring capabilities could improve the transparency, accountability, and impact of RSPO and other certification efforts. RSPO certification prohibits specific categories of forest clearing that cannot be readily distinguished using satellite data. For example, total forest loss can be identified using freely available satellite data products, but HCV or primary forest types cannot be confirmed with Landsat or MODIS data. Changing RSPO criteria to more closely match existing satellite data products on forest cover and forest loss would enable more rigorous monitoring of environmental compliance. Our study identified forest loss on plantations seeking certification after the cut-off date set by RSPO, but observations of forest loss do not confirm non-compliance with the Principles &

Criteria of certification because HCV and primary forest areas cannot be confirmed using satellite data. Alternatively, public databases of set-aside areas on certified plantations (e.g., stream buffers, areas deemed unsuitable for production, or HCV) could improve transparency and support monitoring efforts without the need to derive forest conditions directly from satellite data. New, higher resolution active fire data also complement the time series of MODIS active fire observations. Landsat

and VIIRS active fire data offer sufficient spatial detail to unambiguously attribute fire activity to specific land owners—an important step forward in satellite monitoring by governments, non-governmental organizations, or certification bodies such as RSPO. Fire suppression is important to safeguard carbon stocks in peatlands, and Landsat resolution is particularly beneficial for early detection of new wildfires and to identify small, smouldering fires in peat areas (Schroeder et al., 2015; Elvidge et al., 2015) that may persist for weeks and exacerbate GHG emissions and regional air quality during drought

events.

By 2020, Indonesia has pledged to double its palm oil production (Maulia, 2010), and expanding production threatens remaining rainforest and peatland areas. Certification offers a path for low-carbon development of additional oil palm production, provided that certification standards are consistent with capabilities for routine satellite monitoring. RSPO certification has reduced but not eliminated forest loss and fire use on certified plantations. To realize the full potential of

15 certification, requirements for RSPO certification must be updated to align environmental goals with objective measures of compliance, including industry commitments to a goal of zero deforestation. Such transparency would also provide more direct insight into the key mechanisms through which agricultural intensification and expansion contribute to feedbacks in the Earth system.

**Acknowledgements**

Funding for this study was provided by NASA's Carbon Monitoring System and Interdisciplinary Science Programs and the Norwegian Agency for Development Cooperation's Civil Society Department under Norway's International Climate and Forest Initiative (NORAD, Grants QZA-0465 and QZA-13/0075).

**Appendix A Figures**

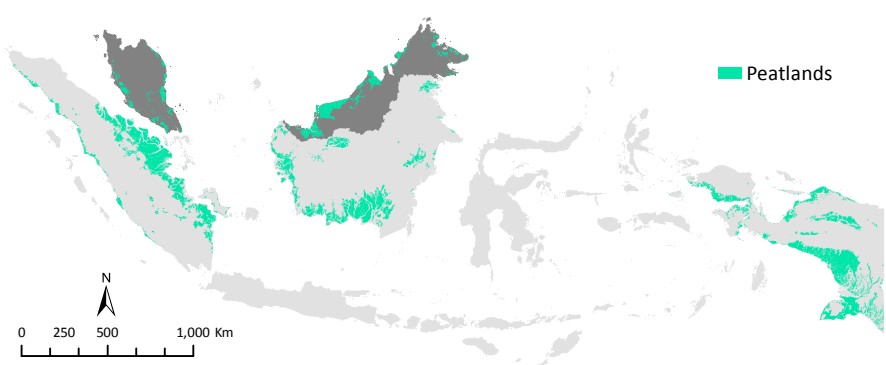

Figure A1: Extent of peatland in Indonesia and Malaysia (Wahyunto et al., 2003; 2004;2006 and WI, 2016).

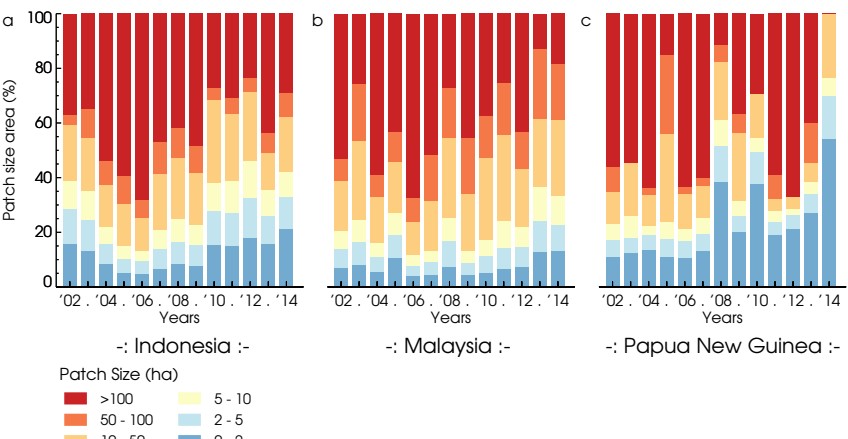

Figure A2: Forest loss patch size distribution in the RSPO Certified plantations of a) Indonesia, b) Malaysia, and c) Papua New Guinea. Patch sizes were assessed at the plantation level and summarized annually based on the proportion of total forest loss in
10  each size class during 2002-2014.

Praveen Noojipady 7/14/2017 4:16 PM
**Deleted:** A5

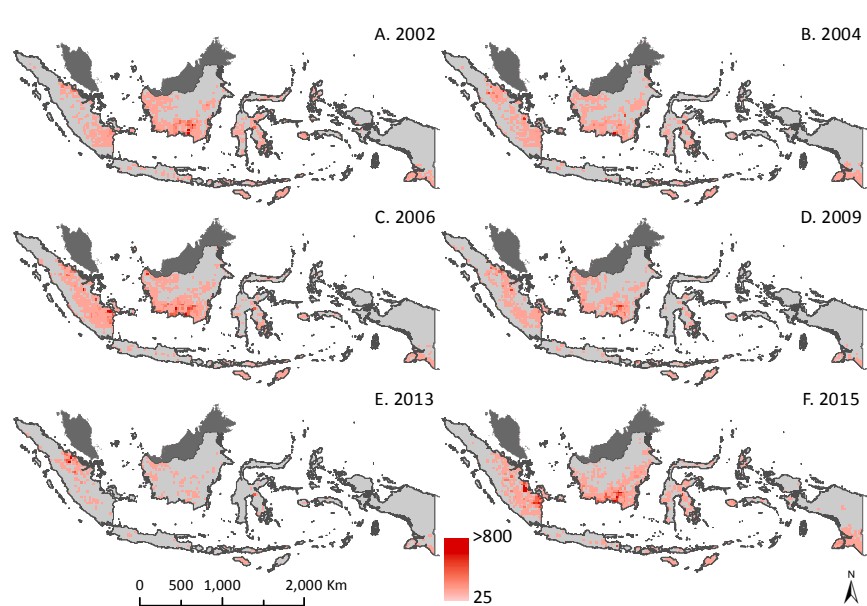

Figure A3: Density of MODIS active fire detections in Indonesia during El Niño years (A-D, F) and the June 2013 drought (E), when fires from Sumatra impacted air quality in Singapore (Gaveau et al., 2014). The spatial distribution of fire activity was consistent during El Niño years, although fire densities were highest in 2006 and 2015. Maps show annual totals of Terra and Aqua MODIS fire detections at 0.25° resolution.

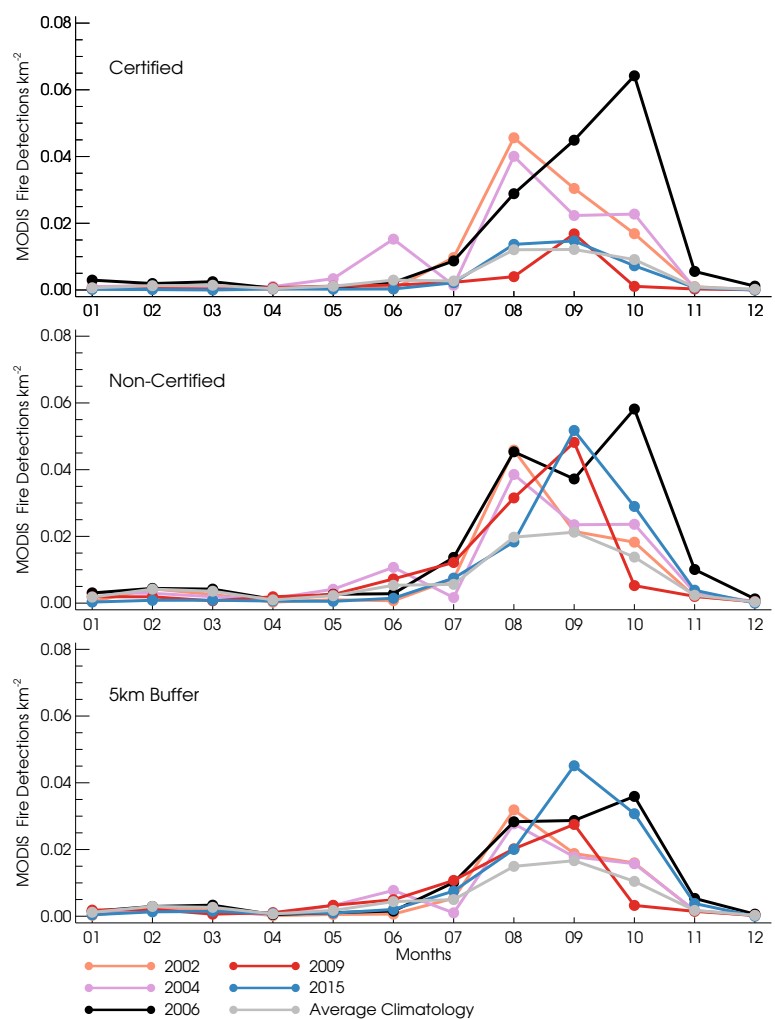

**Figure A4: Monthly density of MODIS active fire detections (Terra and Aqua, combined) for certified plantations, non-certified plantations, and a 5-km buffer region surrounding plantations in Indonesia during El Niño years. A climatology of average monthly fire detections from all years (2001-2015) is shown in grey.**

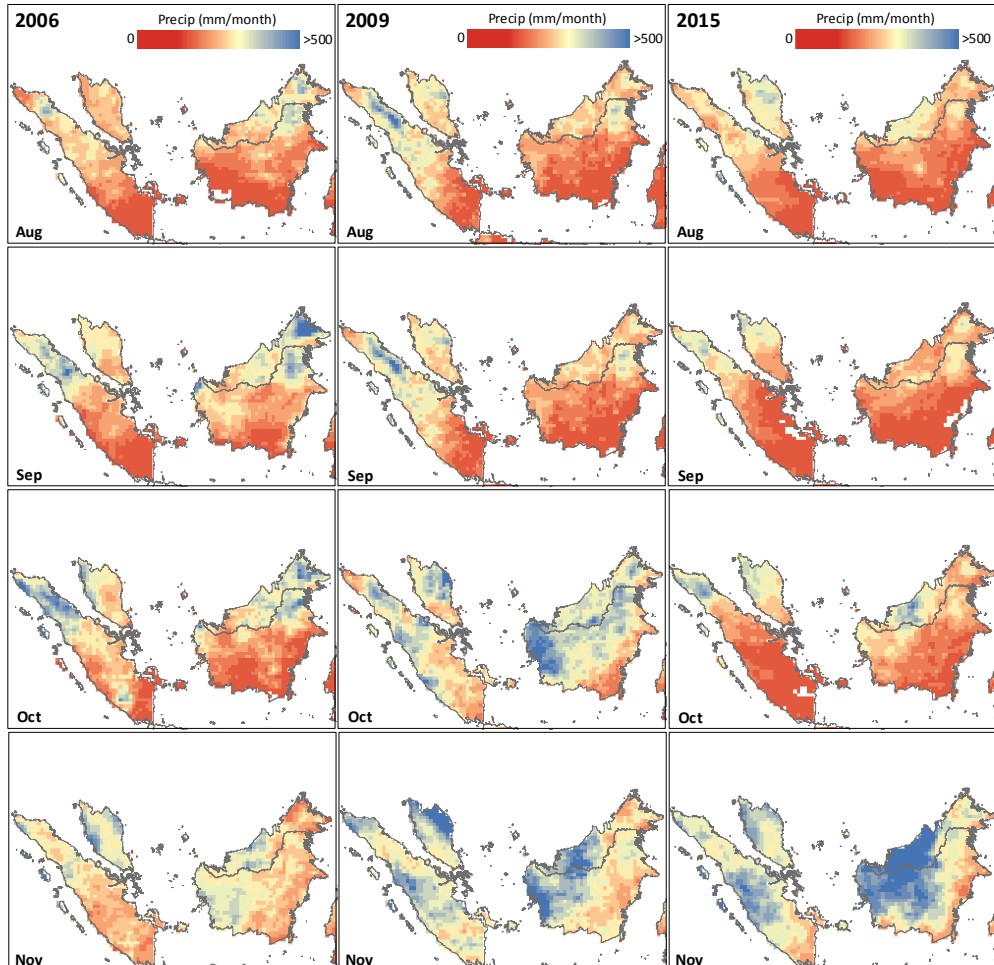

Figure A5: Monthly precipitation patterns for Indonesia and Malaysia from Tropical Rainfall Measuring Mission (TRMM) at 0.25° resolution for months with peak fire activity during the 2006, 2009, and 2015 El Niño events. Certified and non-certified plantations are clustered in similar locations (see Fig. 1); 73% of certified plantations were directly adjacent to one or more non-certified plantations, and 89% of certified plantations were within 10 km of a non-certified plantation. Given this

clustering, and the spatial resolution of precipitation estimates from the TRMM satellite, we assume that precipitation reductions during El Niño events influence certified and non-certified plantations in a similar fashion.

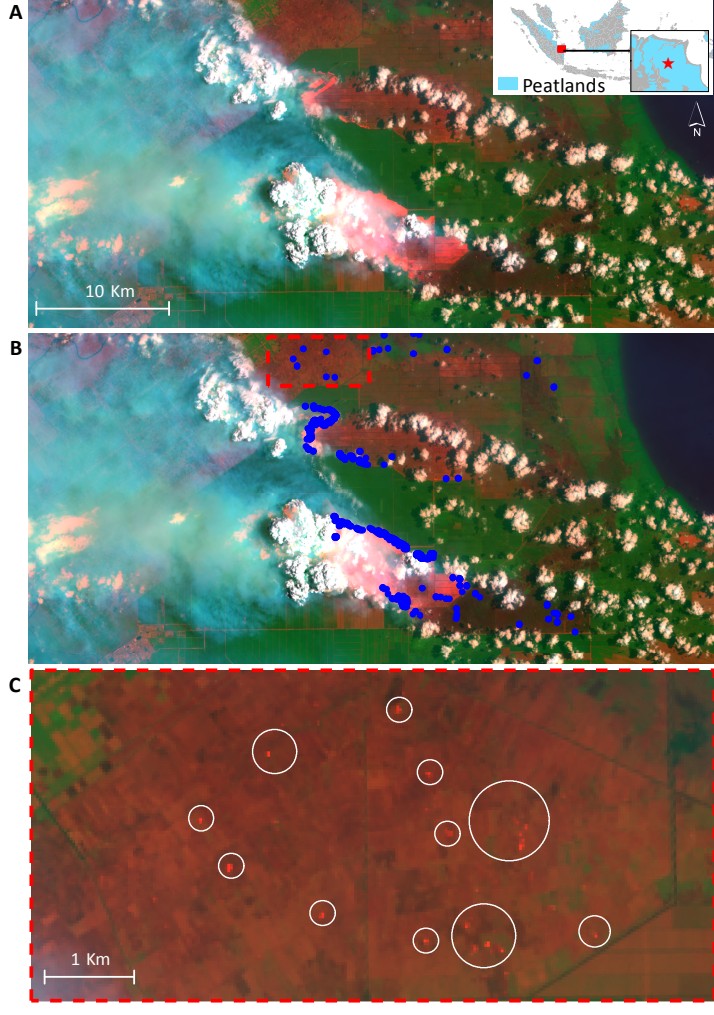

**Figure A6**: Landsat 8 active fire detections captured active fire fronts (B) and residual smouldering fires (C) in peatland areas of southern Sumatra on Sep. 30, 2015. White circles in panel C indicate smouldering for a subset of the image in panel B (dashed red outline). The regular grid of peatland drainage canals is visible in all panels.

5 **Appendix B Tables**

**Table B1**: Total and fire-driven forest loss for oil palm expansion in certified plantations in Indonesia, Malaysia, and Papua New Guinea during 2002-2014. See Tables B2 – B5 for annual estimates of total forest loss, fire-driven forest loss, and the proportion of forest losses on plantations after the receipt of RSPO certification. All areas are given in hectares (ha).

|  | Lease area (ha) | Planted palm by 2010[a] (ha) | Forest loss (ha) | Peat Forest loss (ha) | Fire-driven loss[b] (ha) |
|---|---|---|---|---|---|
| Indonesia (IDN) | 1,652,644 | 1,212,669 | 267,931 | 61,750 | 85,524 (26%) |
| Malaysia (MYS) | 1,113,686 | 877,629 | 121,958 | 5,931 | 5,925 (5%) |
| Papua New Guinea (PNG) | 174,439 | 94,001 | 21,491 | - | 3,860 (18%) |

[a] Forest loss outside of peat areas

10 [b] Combined (peat and non-peat) forest loss related to fire

**Table B2**: Total and fire-driven forest loss for oil palm expansion in Indonesia from 2002-2014 within the certified and non-certified plantations. The percentage of total forest loss on plantations with RSPO certification is shown beginning in 2009. Totals exclude peat forest loss (see Table B3).

| Year | Certified | | | | Non-Certified | | Buffer 5km | |
|---|---|---|---|---|---|---|---|---|
| | Total loss (ha) | Post-Certification loss (%) | Fire-driven loss (ha) | Post-Certification loss (%) | Total loss (ha) | Fire-driven loss (ha) | Total loss (ha) | Fire-driven loss (ha) |
| 2002 | 12,646 | | 4,961 | | 86,179 | 21,890 | 184,140 | 29,713 |
| 2003 | 7,043 | | 2,552 | | 53,578 | 18,693 | 104,882 | 23,135 |
| 2004 | 32,885 | | 12,587 | | 158,904 | 62,232 | 288,634 | 71,538 |
| 2005 | 33,795 | | 9,170 | | 140,345 | 42,260 | 244,178 | 56,281 |
| 2006 | 54,313 | | 12,023 | | 224,249 | 85,081 | 320,690 | 88,869 |
| 2007 | 34,218 | | 6,905 | | 203,990 | 61,875 | 303,782 | 67,606 |
| 2008 | 27,376 | | 876 | | 252,538 | 31,337 | 355,449 | 47,793 |

| 2009 | 29,229 | (1) | 2,543 | (0) | 335,246 | 62,356 | 446,635 | 79,842 |
|------|--------|-----|-------|-----|---------|--------|---------|--------|
| 2010 | 6,267 | (8) | 306 | (0) | 120,598 | 14,330 | 228,111 | 28,634 |
| 2011 | 7,105 | (23) | 308 | (42) | 240,864 | 22,776 | 316,644 | 34,771 |
| 2012 | 9,163 | (25) | 495 | (25) | 334,453 | 45,787 | 512,886 | 80,585 |
| 2013 | 6,628 | (50) | 480 | (82) | 176,080 | 21,815 | 245,738 | 32,635 |
| 2014 | 7,264 | (82) | 774 | (96) | 195,885 | 31,298 | 302,012 | 48,848 |

**Table B3: Total and fire-driven peat forest loss for oil palm expansion in Indonesia from 2002-2014 within certified and non-certified plantations. The percentage of total peat forest loss on plantations with RSPO certification is shown beginning in 2009, the year plantations in Indonesia were first granted RSPO certificates. See Table B2 for lowland forest loss on mineral soils.**

| Year | Certified | | | | Non-Certified | | Buffer 5km | |
|------|-----------|----------------------|-------------|----------------------|----------------|-------------|------------|-----------|
| | Total loss (ha) | Post-Certification loss (%) | Fire-driven loss (ha) | Post-Certification loss (%) | Total loss (ha) | Fire-driven loss (ha) | Total loss (ha) | Fire-driven loss (ha) |
| 2002 | 3,408 | | 452 | | 26,271 | 13,496 | 36,719 | 14,116 |
| 2003 | 1,696 | | 1,007 | | 17,486 | 10,393 | 20,140 | 10,307 |
| 2004 | 6,555 | | 3,789 | | 47,330 | 23,756 | 71,975 | 30,891 |
| 2005 | 7,375 | | 4,514 | | 56,080 | 29,196 | 91,955 | 45,933 |
| 2006 | 18,119 | | 4,938 | | 45,606 | 28,978 | 60,686 | 31,992 |
| 2007 | 9,622 | | 2,741 | | 63,461 | 32,638 | 99,661 | 47,172 |
| 2008 | 4,888 | | 350 | | 60,468 | 13,057 | 65,747 | 16,117 |
| 2009 | 6,632 | (0) | 569 | (0) | 113,140 | 44,731 | 95,858 | 39,048 |
| 2010 | 514 | (3) | 121 | (0) | 38,743 | 11,717 | 70,823 | 23,902 |
| 2011 | 1,429 | (55) | 208 | (89) | 62,348 | 15,796 | 71,104 | 20,570 |
| 2012 | 584 | (64) | 80 | (90) | 100,402 | 30,024 | 120,737 | 46,583 |
| 2013 | 274 | (55) | 19 | (93) | 50,290 | 11,431 | 52,496 | 18,749 |
| 2014 | 654 | (83) | 201 | (96) | 60,400 | 23,404 | 79,305 | 38,035 |

Table B4: Number of certified and non-certified plantations with fire-driven deforestation between 2002-2014. Plantations with fire-driven deforestation after receiving RSPO certification are shown in parenthesis beginning in 2009.

|  | Certified | | | Non-Certified |
|---|---|---|---|---|
| Year | Indonesia N=154 | Malaysia N=119 | Papua New Guinea N=10 | Indonesia N=1536 |
| 2002 | 63 | 20 | 4 | 747 |
| 2003 | 64 | 12 | 5 | 733 |
| 2004 | 82 | 16 | 4 | 913 |
| 2005 | 78 | 18 | 3 | 859 |
| 2006 | 67 | 12 | 5 | 927 |
| 2007 | 66 | 5 | 5 | 902 |
| 2008 | 39 | 8 | 5 | 724 |
| 2009 | 50 (0) | 7 (0) | 3 (0) | 886 |
| 2010 | 35 (1) | 10 (2) | 3 (1) | 738 |
| 2011 | 34 (6) | 12 (5) | 2 (1) | 697 |
| 2012 | 36 (12) | 9 (5) | 2 (1) | 783 |
| 2013 | 39 (17) | 8 (4) | 1 (1) | 692 |
| 2014 | 37 (25) | 8 (6) | 2 (2) | 766 |

Table B5: Total and fire-driven forest loss for oil palm expansion in certified plantations in Malaysia and Papua New Guinea during 2002-2014. All areas are given in hectares (ha), and totals exclude peat forest loss (see Table B5). The percentage of total forest loss on plantations with RSPO certification is shown beginning in 2008.

Microsoft Office User 7/18/2017 5:01 PM
Deleted: 4

| Year | Malaysia | | | | Papua New Guinea | | | |
|---|---|---|---|---|---|---|---|---|
|  | Total loss (ha) | Post-Certification loss (%) | Fire-driven loss (ha) | Post-Certification loss (%) | Total loss (ha) | Post-Certification loss (%) | Fire-driven loss (ha) | Post-Certification loss (%) |
| 2002 | 14,870 | | 912 | | 3,959 | | 1,244 | |
| 2003 | 6,563 | | 791 | | 1,645 | | 301 | |
| 2004 | 13,522 | | 1,912 | | 3,279 | | 721 | |

| Year | | | | | | | | |
|---|---|---|---|---|---|---|---|---|
| 2005 | 6,410 | | 506 | | 1,242 | | 252 | |
| 2006 | 12,312 | | 465 | | 2,893 | | 718 | |
| 2007 | 12,045 | | 15 | | 2,099 | | 479 | |
| 2008 | 7,381 | (2) | 91 | (0) | 1,188 | (34) | 116 | (7) |
| 2009 | 15,467 | (8) | 69 | (0) | 938 | (71) | 3 | (0) |
| 2010 | 10,378 | (19) | 155 | (8) | 716 | (85) | 14 | (96) |
| 2011 | 8,222 | (35) | 120 | (65) | 1,065 | (85) | 4 | (98) |
| 2012 | 7,432 | (48) | 235 | (63) | 1,235 | (79) | 3 | (77) |
| 2013 | 3,261 | (50) | 85 | (78) | 756 | (100) | 0 | (100) |
| 2014 | 4,096 | (82) | 114 | (81) | 477 | (100) | 3 | (100) |

**Table B6: Total and fire-driven peat forest loss for oil palm expansion in certified plantations in Malaysia during 2002-2014. All areas are given in hectares (ha). The percentage of total peat forest loss on plantations with RSPO certification is shown beginning in 2008. See Table B4 for lowland forest loss on mineral soils.**

| Year | Malaysia | | | |
|---|---|---|---|---|
| | Total loss (ha) | Post-Certification loss (%) | Fire-driven loss (ha) | Post-Certification loss (%) |
| 2002 | 303 | | 2 | |
| 2003 | 97 | | 26 | |
| 2004 | 272 | | 0 | |
| 2005 | 388 | | 32 | |
| 2006 | 210 | | 12 | |
| 2007 | 238 | | 0 | |
| 2008 | 300 | (5) | 0 | (0) |
| 2009 | 780 | (37) | 81 | (0) |
| 2010 | 1,146 | (33) | 12 | (1) |
| 2011 | 559 | (35) | 15 | (95) |
| 2012 | 1,243 | (48) | 1 | (0) |
| 2013 | 207 | (58) | 8 | (100) |
| 2014 | 187 | (91) | 7 | (100) |

**Table B7: Total MODIS fire detections for certified plantations in Indonesia, Malaysia, and Papua New Guinea. The percentage of fire detections on plantations with RSPO certification is shown for 2008 – 2015.**

| Year | Indonesia | | Malaysia | | Papua New Guinea | |
|---|---|---|---|---|---|---|
| | Total fire detections | Post-Certification fire detection (%) | Total fire detections | Post-Certification fire detections (%) | Total fire detections | Post-Certification fire detections (%) |
| 2001 | 169 | | 124 | | 37 | |
| 2002 | 1782 | | 87 | | 130 | |
| 2003 | 716 | | 71 | | 64 | |
| 2004 | 1821 | | 87 | | 130 | |
| 2005 | 1008 | | 128 | | 39 | |
| 2006 | 2712 | | 17 | | 83 | |
| 2007 | 197 | | 12 | | 61 | |
| 2008 | 87 | | 9 | (0) | 43 | (7) |
| 2009 | 483 | (0) | 22 | (0) | 31 | (0) |
| 2010 | 72 | (8) | 18 | (28) | 44 | (95) |
| 2011 | 196 | (29) | 12 | (50) | 18 | (67) |
| 2012 | 191 | (39) | 21 | (33) | 44 | (84) |
| 2013 | 128 | (55) | 11 | (55) | 54 | (100) |
| 2014 | 361 | (73) | 35 | (69) | 52 | (100) |
| 2015 | 656 | (100) | 26 | (100) | 136 | (100) |

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

**Figures**

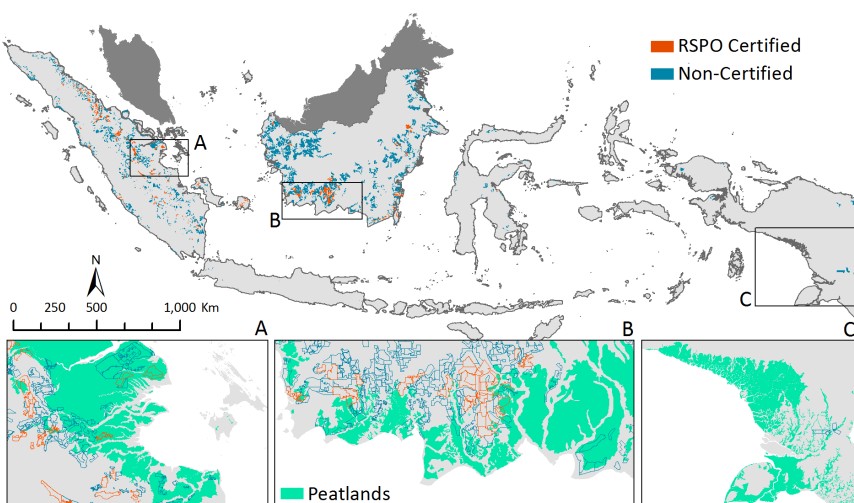

**Figure 1: Extent of RSPO certified (red) and non-certified (blue) oil palm plantations in Indonesia. Regional subsets highlight plantations boundaries on peatlands (green) in lowlands of Sumatra (A), Kalimantan (B), and Papua (C).**

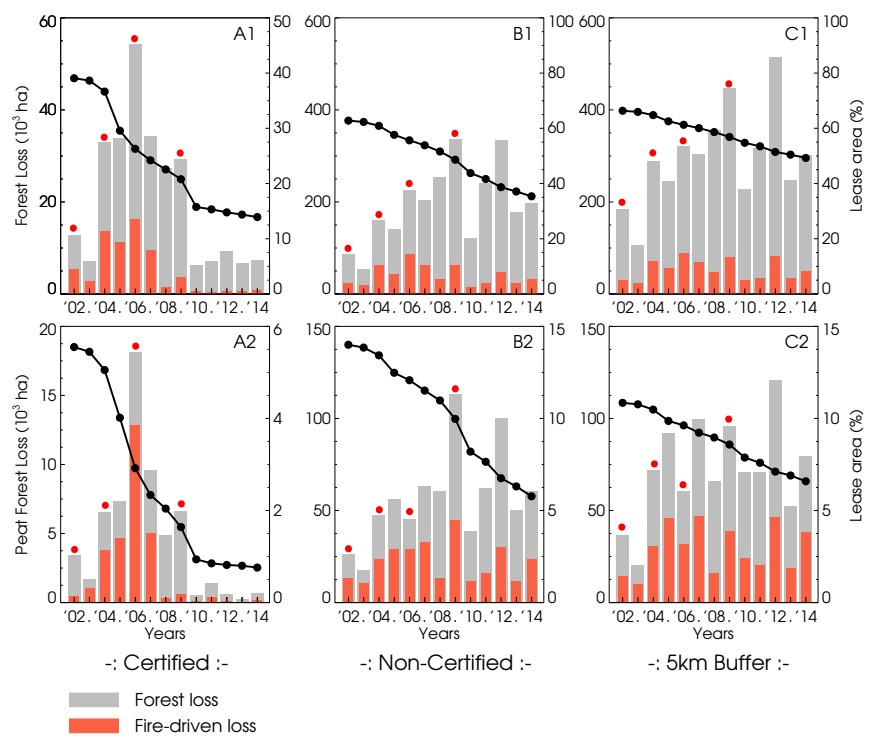

**Figure 2: Forest loss within the boundaries of A) Certified plantations, B) Non-Certified plantations, and C) 5km Buffer region surrounding certified and non-certified plantations in Indonesia from 2002-2014. A1-C1) Forest loss (grey) and fire-driven forest loss (orange); A2-C2) Peat forest loss (grey) and fire-driven peat forest loss (orange). Solid black lines indicate residual forest cover as a percentage of management areas. Annual estimates of total forest loss and fire-driven forest loss are summarized in Tables B2 and B3, including the proportion of forest losses on plantations after the receipt of RSPO certification. Orange dots indicate El Niño years.**

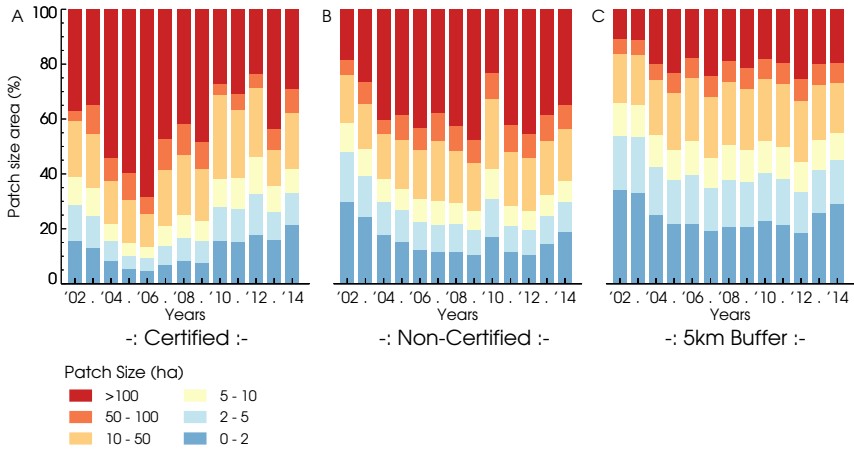

**Figure 3: Forest loss patch size distribution in Indonesia within the boundaries of A) RSPO Certified plantations, B) Non-Certified plantations, and C) 5km Buffer region. Patch sizes were assessed at the plantation level and summarized annually based on the proportion of total forest losses in each size category during 2002-2014.**

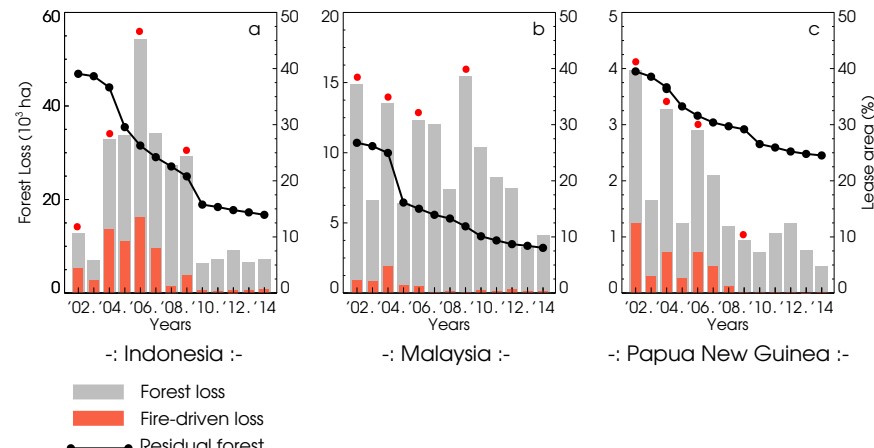

**Figure 4: Total forest loss (grey) and fire-driven forest loss (orange) in certified plantations in a) Indonesia, b) Malaysia, and c) Papua New Guinea. Forest loss was estimated outside of planted palm (Carlson et al., 2013; Gunarso et al., 2013). Black lines indicate remaining forest area as a fraction of the total lease area of certified plantations in each country. Annual estimates of total**

forest loss and fire-driven forest loss for certified plantations are summarized in Tables B2 - B5, including the proportion of forest losses on plantations after the receipt of RSPO certification.  Orange dots indicate El Niño years.

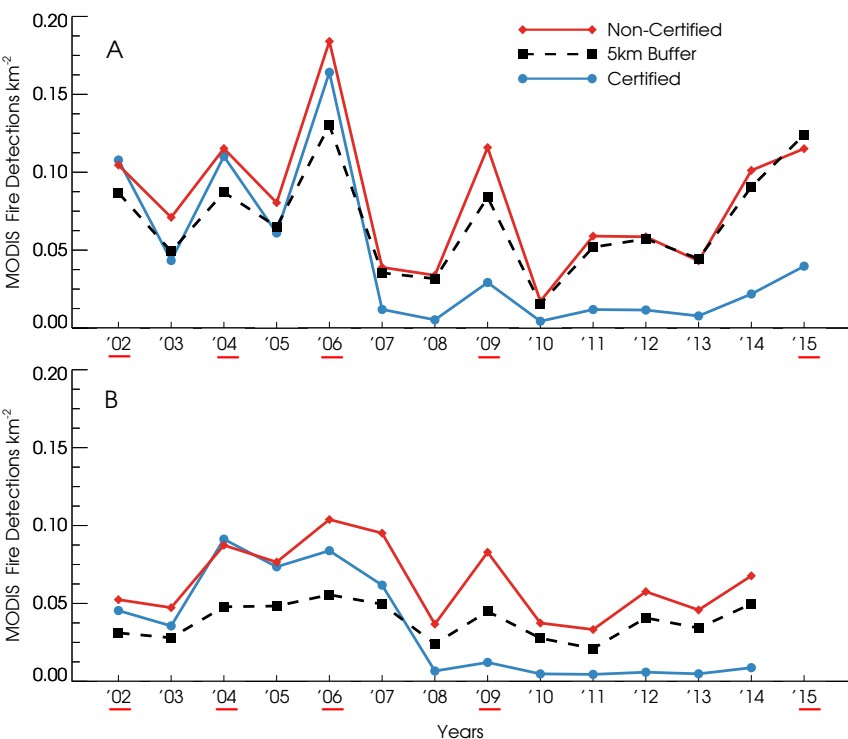

Figure 5: Density of MODIS active fire detections within certified plantations, non-certified plantations, and the 5-km buffer region around plantations in Indonesia from 2002-2014. A) Time series of all MODIS active fire detections; B) Time series of MODIS active fire detections associated with fire-driven deforestation. El Niño years are underlined in red. Table B6 provides annual estimates of fire counts on certified and non-certified plantations, including the proportion of fire detections on plantations after the receipt of RSPO certification.

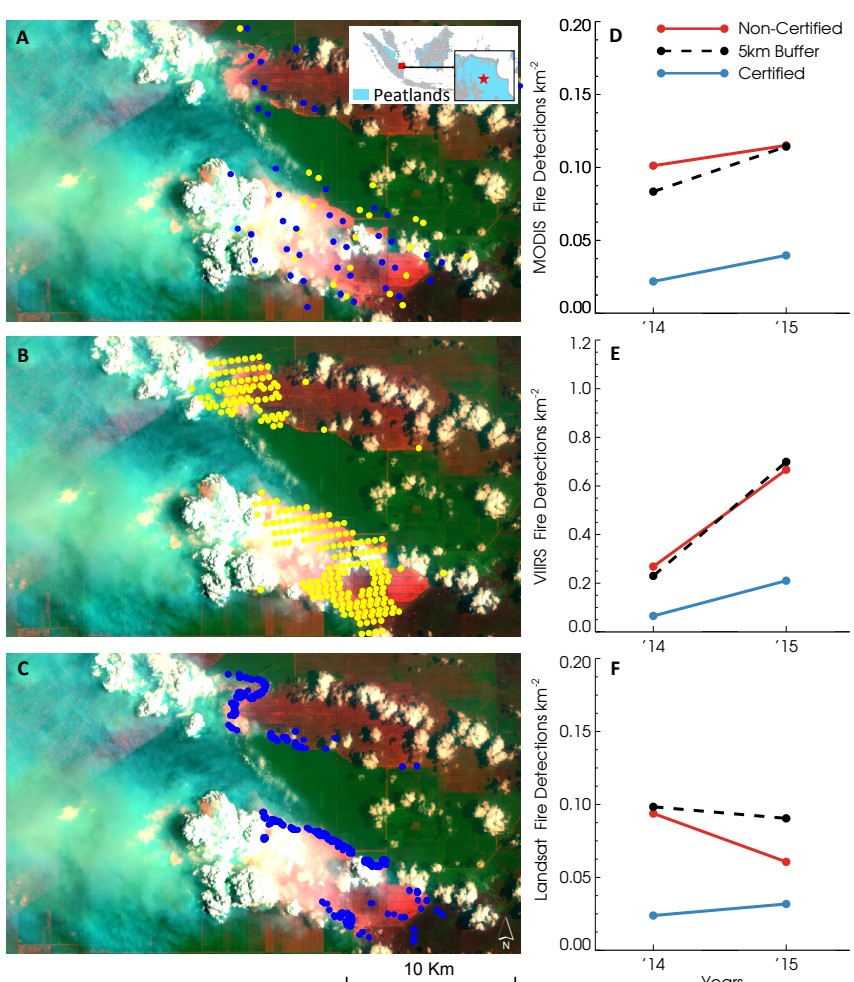

**Figure 6: High-resolution active fire detections confirm lower fire activity in certified plantations during the 2015 El Niño event. Map panels show active fire detections on Sep. 30, 2015 for peat fires in southern Sumatra from A) Terra (blue) and Aqua (yellow) MODIS (1 km), B) S-NPP/VIIRS (375m), and C) Landsat-8/OLI (30m). Background images in panels A-C are a false-color composite of Landsat 8/OLI bands 7-5-3 from the same date (Path/Row: 124/62). Adjacent panels show total annual fire detections in 2014 and 2015 for certified plantations from D) MODIS, E) VIIRS, and F) Landsat 8/OLI.**

**Tables**

**Table 1: Total and fire-driven forest loss for oil palm expansion in Indonesia from 2002-2014 within certified and non-certified plantations. See Tables B2 and B3 for annual estimates of total forest loss, fire-driven forest loss, and the proportion of forest losses on plantations after the receipt of RSPO certification.**

|  | Lease area (ha) | Planted palm by 2010 (ha) | Forest loss[a] (ha) | Peat Forest loss (ha) | Fire-driven loss[b] (ha) |
|---|---|---|---|---|---|
| RSPO Certified | 1,652,644 | 1,212,669 | 267,931 | 61,750 | 85,524 (26%) |
| Non-Certified | 11,266,875 | 3,416,302 | 2,522,909 | 742,025 | 810,346 (25%) |
| 5km Buffer | 25,092,975 | 2,931,778 | 3,853,782 | 937,206 | 1,073,664 (22%) |

[a] Forest loss outside of peat areas

[b] Combined (peat and non-peat) forest loss related to fire

