# Peer review of "Managing fire risk during drought: the influence of certification and El Niño on fire-driven forest conversion for oil palm in Southeast Asia"

_Earth System Dynamics, 2017_

## Referee Comment (RC1) · K. Austin (Referee) · 23 Jan 2017

This is a strong paper which addresses an important question regarding the role of RSPO certification for improving management of fire and fire-driven deforestation in permits for oil palm cultivation. The methods are clear, the report is well written and clearly organized, and the graphics are informative. I have the following questions/comments for the authors:

1. Compliance benchmark date - The paper refers to 2009 as the year that RSPO began granting certification, and notes that forest loss and fire-driven deforestation declined after this date. However, the benchmark date used to determine compliance with RSPO criterion 7.3, after which new plantings should not replace primary forest or

HCV areas, is November 2005. Why does the study use 2009 to assess compliance, given that RSPO uses an earlier year to assess compliance?

Due to the importance of being able to compare pre- and post- certification trends, I would find it useful to present the proportion of forest loss driven by fire each year in a table. It is difficult to see proportions in Figure 2 for years with low rates of forest loss, and Table 1 only provides this the aggregate proportion over the 2000 – 2014 period. Breakdown by year would help illustrate whether, and when, certification alters fire use for forest conversion.

2. Buffer areas - The authors should articulate the purpose of the 5 km buffer area, and in particular clarify why they combine the buffer areas of certified and non-certified plantations. Given that the study assesses roughly 12 Mha of non-certified plantations, versus 1.5 Mha of certified plantations, the trends in the combined buffer will largely reflect the characteristics of buffers around non-certified plantations and presumably more closely resemble the trends inside non-certified plantations. Combining the buffers masks potentially divergent trends in the buffer of the certified plantation management type, and obscures whether certification additionally impacts fire activity in areas surrounding the permit itself.

3. Underestimation of fire activity – The authors discuss limitations in satellite platforms which detect fires, and suggest that these limitations may result in underestimation of fire activity. Is there any reason to think that this underestimation would bias the results, either by differentially underestimating fire density in time (e.g., after certification date), or in space (e.g., in certified concessions)?

4. Covariates of certification – The study could elaborate on the factors which could cause different observed fire and fire driven forest loss trends between certified and non-certified plantations. The authors mention that companies preferentially certify older plantations that retain less forest cover. If this is the case, or if there are other characteristics which influence the placement of certified plantations or the outcomes

with respect to forest loss and fire activity in certified plantations, then observed differences may not be the result of certification. The authors should clearly caveat the findings by acknowledging these covariates, and/or suggest what steps would be necessary to control for these in order to determine the causal impacts of certification.

5. Policy implications – The authors could further elaborate on the policy implications of their findings. They suggest that the benefits conferred by RSPO certification could be enhanced through expansion of certified plantations. How would this work? Given that certification is based on performance after the benchmark date, many plantations will poor past performance may not be eligible for certification (or would need to take advantage of a compensation mechanism). Would the expansion of the certified plantation portfolio therefore only apply to new plantations?
* * *

---

## Referee Comment (RC2) · Anonymous Referee #2 · 10 Feb 2017

General Comments:

-I would like to see more information on how the area for the buffer analysis was selected. Why were the buffer areas around certified and non-certified plantations combined together? Or would it have made more sense to consider the plantation boundaries vs. plantations+buffers, while also keeping certified and non-certified separate? I'm not sure if you might expect differences in fire activity between buffers around each type of plantation. Please also see specific comments below on this topic.

-Could there be differences in characteristics besides certification that are influencing the results? It's not clear to me as written if the authors considered other potential variables such as the level of access to plantations, size, whether part of the concession

[Figure]

was previously developed, differences in specific provinces, etc. This might also help to address the statistical significance of the results.

-Can the authors clarify in the text when they are discussing fires within a year of deforestation (fire-driven deforestation) vs. fires for plantation management/escaped fires? Sometimes it's not clear to me which fire type is being discussed and the description in the methods section does not make this aspect clear.

Specific Comments:

-Pg. 2, Line 24: What about the % certified within Southeast Asia?

-Pg. 3, Line 31: Do you have the date of certification for each plantation or is it only known to have occurred between 2009-2015?

-Pg. 4, Line 4: Was each individual plantation owned by a separate company, or was there overlap in ownership?

-Pg. 4, Line 10: Can you give more details on how planted oil palm was detected and if there were any differences between the three studies?

-Pg. 4, Line 29: How was the 5km buffer selected? Were any differences considered between small vs. large plantations?

-Pg. 5, Line 12: Could there be any effects of having a 5 year timestep for the oil palm datasets vs. the annual deforestation datasets?

-Pg. 5, Line 23: Can you clarify if the certification timing was similar for all of these plantations (2009?) or if it varied across the study area? Could some of the plantations in the certified category have only been certified towards the end of the study period? If the dates are not known, I would appreciate a discussion at some point in the paper on how this could impact results.

-Pg. 5, Lines 23-24: Can you comment here or in the discussion on why this could be higher? Were these plantations easier to access or were there other factors that lead

to higher deforestation pre-certification? Are these results statistically significant?

-Pg 6, Line 1: What do you mean by management classes? Certified, non-certified, and buffers?

-Pg. 6, Line 1: Can you mark el nino years on the figure for reference? Any differences depending on the strength of the el nino?

-Pg. 6, Line 6: Were the number of dry years consistent between the two periods of comparison?

-Pg. 6, Line 15: Again, I'm wondering if you know about differences in certification timing among the three areas?

-Pg. 6: Line 28: Can you give a comparison of the strength of these different El Nino events?

-Pg. 6, Line 35: I'm not sure I understand exactly what you did here. For the annual fire detections, did you address the difference in temporal sampling between the different datasets? What detection differences might you expect between the different sensors and how could this influence comparisons?

-Pg. 7, Line 10: I thought that the Cattau study was focused on concessions that were previously cleared or planted, so wouldn't you expect differences between that study vs. fires used for deforestation as examined here? Or are you considering management fires (see general comment #3)? Not sure if I'm missing something here, so a clarification would be appreciated.

-Discussion: If you feel it's warranted, could you comment on whether your work relates to the findings by Gaveau et al. (2016) on the timing of deforestation for oil palm plantations?

Gaveau, D. L. A. et al. Rapid conversions and avoided deforestation: examining four decades of industrial plantation expansion in Borneo. Sci. Rep. 1–13 (2016).

doi:10.1038/srep32017

Technical Comments:

-Pg. 3, Line 25: Should it be section 2.1? (Also the rest of the subheadings in this section)

-Pg. 5, Line 1: The VIIRS definition just repeats the first part of the sentence?

-Pg, 5: Line 14: Can you add a supplementary figure show the distribution of peatlands? We only have the subsets from Figure 1.

-Pg. 5, Line 22: Missing %.

-Pg. 6, Line 29: What were the peak burning months?

-Figure 1: Is it possible to color code the zoomed in subsets by certified vs. non certified? Perhaps with some shading of the peatlands instead? This might make the figure too busy but it would be nice to see the spatial details.
* * *

---

## Referee Comment (RC3) · Anonymous Referee #3 · 17 Feb 2017

The authors compare fire activity and deforestation between RSPO-certified and non-certified oil palm plantations in Southeast Asia, arguing that RSPO certification has led to reduced fire activity during dry years. This is a well-written paper and the overall result is important. My only significant concern is the assumption that dry conditions during the big fire years were the same with respect to the locations of certified and non-certified plantations.

Comments P1L21: should this be 'did not stop altogether'?

P3L30: I didn't understand the '(ever)' and '(never)' wrt certified and non-certified

P4L20: The end of this sentence implies that Southeast Asia has little rainfall seasonality, which I don't think you mean to say.

P6L28: How are you excluding the possibility that the certified plantations just weren't as dry in 2015 compared to, say, 2006? Figures A3 and Figure 5 clearly show a drop in fire activity over the analysis period over the certified plantations, but from Figure 1, these plantations are not evenly distributed across Sumatra and Kalimantan. It's possible that these regions, for example south-central Kalimantan, were just wetter in 2015 than previous years, given that regional rainfall can vary across El Niño years. Or perhaps they were drier, in which case your argument about RSPO effects is strengthened. Either way, regional rainfall needs to be looked at or mentioned as a possible factor.

P7L9: change 'direct' to 'directly'

---

## Author Comment (AC1) · 29 Mar 2017

This is a strong paper which addresses an important question regarding the role of RSPO certification for improving management of fire and fire-driven deforestation in permits for oil palm cultivation. The methods are clear, the report is well written and clearly organized, and the graphics are informative.

We appreciate the reviewer's recognition of the importance of our study on fire-driven deforestation in Southeast Asia and the role of RSPO certification.

I have the following questions/ comments for the authors:

1. Compliance benchmark date - The paper refers to 2009 as the year that RSPO

began granting certification, and notes that forest loss and fire-driven deforestation declined after this date. However, the benchmark date used to determine compliance with RSPO criterion 7.3, after which new plantings should not replace primary forest or HCV areas, is November 2005. Why does the study use 2009 to assess compliance, given that RSPO uses an earlier year to assess compliance?

We agree with the reviewer that there are several important dates to consider regarding the evolution of RSPO and related criteria for certified plantations.

While the RSPO standard requires no replacement of primary forests or HCV areas after November 2005, the incentives to follow this requirement, and compliance with the requirement, have changed over time. The RSPO draft standard was piloted for two years (2005-2007) with volunteer oil palm companies who committed to avoid clearing primary intact forest and High Conservation Value (HCV) areas. Although this initial standard was approved in 2007, the first RSPO certificates for sustainable oil palm in Malaysia and Indonesia were not issued until 2008 and 2009, respectively. In 2010, the New Planting Procedure required that all member companies conduct an HCV assessment prior to clearing. In 2014, the Remediation and Compensation procedure recognized that RSPO members had cleared and planted after 2005 without first conducting an HCV assessment, and required companies to compensate for such clearance.

In a revised manuscript, we would clarify the evolution of the RSPO standard and additional requirements.

Specifically, we would discuss the sequence of events that precede certification in the Introduction, including company membership in RSPO and the agreement of all members to follow the Principles & Criteria, including Criterion 7.3:

"Companies interested in certification first become members of the RSPO and agree to the Principles & Criteria, including the prohibition on new plantings through deforestation of primary or HCV forests after Nov. 2005 without compensation (Criterion 7.3, RSPO 2013)."

In the methods, we propose to clarify the key dates for our evaluation of fire-driven deforestation and total fire activity: "Comparisons between certified (ever) and non-certified (never) plantations considered forest loss and fire activity over three time scales: 1) following the benchmark date for compliance with RSPO criterion 7.3 (Nov. 2005), 2) following the first issuance of RSPO certificates to Indonesian producers in 2009, and 3) following the date of certification for individual plantations."

In the results section, we propose to separately consider deforestation and fire activity during 2006-2009, 2009-2015, and the subset of all deforestation and fire activity on plantations with RSPO certification. For example: "In Indonesia, annual rates of forest loss and fire-driven forest loss were higher in certified plantations before the first RSPO certificates were issued (2006-2008, 38,636 ha yr-1) than during 2009-2015 (10,943 ha yr-1). Between 2009 and 2012, the majority of forest loss and fire-driven forest loss occurred on plantations that had not yet received RSPO certification (Table B2), whereas plantations with certificates accounted for most forest losses identified in 2013-2014."

Due to the importance of being able to compare pre- and post- certification trends, I would find it useful to present the proportion of forest loss driven by fire each year in a table. It is difficult to see proportions in Figure 2 for years with low rates of forest loss, and Table 1 only provides this the aggregate proportion over the 2000 – 2014 period. Breakdown by year would help illustrate whether, and when, certification alters fire use for forest conversion.

Below, we present annual forest loss and fire-driven forest loss for certified, non-certified, and buffer areas in Indonesia (Table B2) and certified plantations in Malaysia and Papua New Guinea (Table B3). In a revised manuscript, these tables could be presented online to complement the material in Figure 2 and Figure 4, or added to the main text for completeness. In addition, we have estimated the proportion of total forest loss and fire-driven forest loss on plantations following the receipt of RSPO certification, noting that the first RSPO certificates were issued to plantations in Indonesia

in 2009 and in 2008 to plantations in Malaysia and Papua New Guinea. Table B4 provides a similar breakdown of total fire detections for certified plantations, including post-certification MODIS fire detections.

\*\*\*\*\*\*\*\*\*\*\*\*\*\*\*\*\*\*\*\*\*\*\*\*\*\*\*\*\*\*\*\*\*\*\*\*\*\*\*\*\*\*\*\*\*\*\*\*\*\*\*\*\*

2. Buffer areas - The authors should articulate the purpose of the 5 km buffer area, and in particular clarify why they combine the buffer areas of certified and non-certified plantations. Given that the study assesses roughly 12 Mha of non-certified plantations, versus 1.5 Mha of certified plantations, the trends in the combined buffer will largely reflect the characteristics of buffers around non-certified plantations and presumably more closely resemble the trends inside non-certified plantations. Combining the buffers masks potentially divergent trends in the buffer of the certified plantation management type, and obscures whether certification additionally impacts fire activity in areas surrounding the permit itself.

We appreciate the reviewer's suggestion to clarify our analysis of buffer areas surrounding oil palm plantations. Human ignitions are the dominant source of fire activity in Southeast Asia, and recent fire emergencies (e.g., 2013 and strong El Niño events in 1997-1998, 2002, 2006, and 2015) have intensified the debate over the source of fire ignitions. Two questions guided our inclusion of buffer areas surrounding plantations: 1) what is the role of smallholders adjacent to oil palm plantations for total fire activity observed from satellite sensors? 2) do buffer areas exhibit similar patterns of interannual variability in fire-driven forest clearing and fire detections as oil palm plantations? In other words, our main goal was to characterize fire and deforestation immediately surrounding oil palm plantations, not to contrast certified and non-certified buffers.

Buffer areas surrounding non-certified plantations cover a larger land area, as the reviewer correctly points out, but the patterns of remaining forested area and forest loss are similar for buffer areas surrounding certified and non-certified plantations (Figure R1). In addition, oil palm plantations in Southeast Asia are frequently adjacent to other

oil palm plantations (Figure 1, see also response to Reviewer #3), meaning that it is difficult to attribute buffer activities to only certified or non-certified neighbors. As a result, we analyzed fire activity and forest loss for a single set of buffer areas surrounding certified and non-certified plantations. The 5km buffer was chosen based on expert judgment to capture the potential influence of forest loss and fire activity on the surrounding landscape, including the direct fire spread from adjacent lands into palm oil plantations, wind-blown embers from nearby fires, and the most acute impacts of smoke on both human health and ecosystems.

In the revised manuscript, we would clarify the characteristics of buffer landscapes in Section 2.1, including the fact that nearly 12% of the area within the 5km buffer was mapped as planted oil palm in 2010 (Gunarso et al., 2013; Carlson et al., 2013). Thus, the buffer region may reflect differences in management, in addition to differences in land use and land cover, based on the abundance of planted palm oil outside of large plantations.
* * *
3. Underestimation of fire activity – The authors discuss limitations in satellite platforms, which detect fires, and suggest that these limitations may result in underestimation of fire activity. Is there any reason to think that this underestimation would bias the results, either by differentially underestimating fire density in time (e.g., after certification date), or in space (e.g., in certified concessions)?

The long-term record of Moderate Resolution Imaging Spectroradiometer (MODIS) active fire detections (MCD14ML) provides a consistent daily assessment of fire activity during the entire study period. MODIS detections therefore provide clear and consistent evidence for interannual (Figure 5) and monthly (Figure A3) variability in total fire detections in and around oil palm plantations. Limitations of MODIS fire detection from orbital coverage, cloud cover, or spatial resolution are also consistent over time, and are therefore unlikely to bias the analysis of fire activity in specific years or specific

plantations.

The launch of additional satellite platforms, including the Visible and infrared Imaging Radiometer Suite (VIIRS) on the Suomi-National Polar orbiting Partnership (S-NPP) and Operational Land Imager (OLI) on Landsat-8, provide higher spatial resolution active fire information to detect smaller (or cooler) active fires and provide an unambiguous attribution of fire activity to specific land holders (see Figure 6). In the original manuscript, we highlighted the potential to improve operational satellite monitoring of fire activity using these new sensors, specifically to monitor land use change and environmental compliance. However, the benefits of new sensor systems do not diminish the value of long-term monitoring using MODIS. We propose to clarify the importance of the long and consistent MODIS data record in a revised manuscript: "We used the time series of active fire detections from the Moderate Resolution Imaging Spectroradiometer (MODIS) instruments on NASA's Terra and Aqua satellites to evaluate the spatial and temporal patterns of daily fire activity during 2002-2015."

"The finer spatial resolution of these fire data capture additional details regarding fire activity that can be difficult to evaluate at MODIS resolution, including the precise location of active fire fronts, separation of flaming and smoldering fires (Elvidge et al., 2015), and detection of small and/or lower intensity fires (Schroeder et al., 2015)—an important component of fire activity in agricultural landscapes (Randerson et al., 2012)."

The potential for certification or other changes in land use and land management practices to alter the probability of detecting active fires from space is a separate issue from sensor performance. In our original manuscript, we specifically considered one aspect that could influence detection of fire-driven forest loss—changes in the size of clearings over time (Figure 3)—as smaller fires may be more difficult to detect using all satellite platforms. In addition to changes in clearing size, it is possible that land managers have altered the use of fires to avoid detection by burning during cloudy periods when detections are less likely or time intervals with less satellite coverage. New

satellite sensors may partially address issues associated with smaller fires, but not with targeted burning to avoid detection. The potential management response to satellite monitoring capabilities is an interesting direction for further research, but a thorough evaluation of this issue is beyond the scope of this study. In a revised manuscript, we would recognize this line of research as a possible extension of this work using multiple sources of satellite-based fire detections.
* * *
4. Covariates of certification – The study could elaborate on the factors, which could cause different observed fire and fire driven forest loss trends between certified and non-certified plantations. The authors mention that companies preferentially certify older plantations that retain less forest cover. If this is the case, or if there are other characteristics which influence the placement of certified plantations or the outcomes with respect to forest loss and fire activity in certified plantations, then observed differences may not be the result of certification. The authors should clearly caveat the findings by acknowledging these covariates, and/or suggest what steps would be necessary to control for these in order to determine the causal impacts of certification.

We agree with the reviewer that a range of factors will influence fire-driven forest loss on oil palm plantations. In this study, we were primarily interested in the use of fire for forest conversion and interannual variability of fire detections in and around oil palm plantations. For these studies, the larger sample size of all certified and non-certified plantations fills an important data gap in our understanding carbon emissions from oil palm expansionâ the degree of fire use during forest conversion. Companies and consumers purchasing palm oil are concerned about the amount of "embodied" emissions from forest loss and fire activity. Our study quantifies the amount of fire-driven forest loss embodied in certified palm oil based on a more inclusive look at the aggregate behavior among all certified oil palm plantations. As the reviewer points out, key aspects of the RSPO Principles & Criteria would be known to all RSPO members in the process of certifying plantations, including the Nov. 2005 benchmark date for

deforestation of primary or HCV forest lands and the need to comply with environmental legislation banning fire activity.

Previous work that considered a broad range of matching criteria between certified and non-certified plantations was restricted to very small sample sizes (e.g., 4 plantations on peatlands in Cattau et al., 2016). In a separate study, we have more formally controlled for the diversity of plantation characteristics, including remaining forest cover, planted palm oil, age, and date of certification, among others (Carlson et al., under review). In a revised manuscript, we would specifically recognize that our results do not establish causality for differences between certified and non-certified plantations: For example:

"Following the start of RSPO certification in 2009, certified oil palm plantations in Indonesia had lower fire-driven deforestation and total fire activity during El Niño events than non-certified plantations. These reductions point to the potential for RSPO to contribute to REDD+ and to decrease fire ignitions during drought conditions, but our results do not provide conclusive evidence for a causal relationship between certification and lower fire activity."

\*\*\*\*\*\*\*\*\*\*\*\*\*\*\*\*\*\*\*\*\*\*\*\*\*\*\*\*\*\*\*\*\*\*\*\*\*\*\*\*\*\*\*\*\*\*\*\*\*\*\*\*\*\*\*\*\*\*\*\*\*

5. Policy implications – The authors could further elaborate on the policy implications of their findings. They suggest that the benefits conferred by RSPO certification could be enhanced through expansion of certified plantations. How would this work? Given that certification is based on performance after the benchmark date, many plantations will poor past performance may not be eligible for certification (or would need to take advantage of a compensation mechanism). Would the expansion of the certified plantation portfolio therefore only apply to new plantations?

As the reviewer suggests, more widespread adoption of RSPO certification could be hindered by poor past performance because the barriers to entrance are likely to be high for a company that cleared after 2005 without an HCV assessment. However,

entrance is not impossible due to the 2014 RSPO Remediation and Compensation Procedure that would allow certification of existing plantations in non-compliance with the 2005 cut-off date. Under this procedure, RSPO member companies that clear without an HCV assessment after 2014 are expelled from the RSPO, while non-member companies face steep costs to entrance. In the 2005-2014 period, companies face liability based on clearance date and membership status.

Given our findings that certified plantations in Indonesia had significantly lower fire-driven deforestation, even existing non-certified plantations with remaining forest could benefit from improved management practices associated with certification. However, even with full compliance with the RSPO Principles and Criteria, the overall environmental gains of certification may be limited, as the current Principles & Criteria do not restrict the clearance of all forests, only those designated as HCV or "primary."

Importantly, our study highlights how the RSPO Principles & Criteria differ from existing capabilities for remote monitoring of environmental compliance. In a revised manuscript, we would further emphasize the benefits of revised certification criteria that can be more easily monitored using existing satellite sensors. In addition to improving transparency, updating certification criteria to match monitoring capabilities would also bring RSPO more in line with industry commitments to zero-deforestation goals, including the New York Declaration on Forests.

\*\*\*\*\*\*\*\*\*\*\*\*\*\*\*\*\*\*\*\*\*\*\*\*\*\*\*\*\*\*\*\*\*\*\*\*\*\*\*\*\*\*\*\*\*\*\*\*\*\*\*\*\*\*\*\*

References:

Carlson, K. M., Curran, L. M., Asner, G. P., Pittman, A. M., Trigg, S. N. & Marion Adeney, J. 2013. Carbon emissions from forest conversion by Kalimantan oil palm plantations. Nature Clim. Change, 3, 283-287.

Cattau, M. E., Marlier, M. E. & DeFries, R. 2016. Effectiveness of Roundtable on Sustainable Palm Oil (RSPO) for reducing fires on oil palm concessions in Indonesia
from 2012 to 2015. Environmental Research Letters, 11, 105007.

Elvidge, C. D., Zhizhin, M., Hsu, F.-C., Baugh, K., Khomarudin, M. R., Vetrita, Y., Sofan, P. & Hilman, D. 2015. Long-wave infrared identification of smoldering peat fires in Indonesia with nighttime Landsat data. Environmental Research Letters, 10, 065002.

Gunarso, P., Hartoyo, M., Agus, F. & Killeen, T. 2013. Oil palm and land use change in Indonesia, Malaysia and Papua New Guinea. Reports from the Technical Panels of the 2nd greenhouse gas working Group of the Roundtable on Sustainable Palm Oil (RSPO), 29-64.

Randerson, J. T., Chen, Y., van der Werf, G. R., Rogers, B. M. & Morton, D. C. 2012. Global burned area and biomass burning emissions from small fires. Journal of Geophysical Research: Biogeosciences, 117, n/a-n/a.

Schroeder, W., Oliva, P., Giglio, L., Quayle, B., Lorenz, E. & Morelli, F. 2015. Active fire detection using Landsat-8/OLI data. Remote Sensing of Environment.
* * *
[Figure]

Table B2: Total and fire-driven forest loss for oil palm expansion in Indonesia from 2002-2014 within the certified and non-certified plantations.

| Year | Certified | | | | Non-Certified | | Buffer 5km | |
|---|---|---|---|---|---|---|---|---|
| | Total loss (ha) | Post-Certification loss (%) | Fire-driven loss (ha) | Post-Certification loss (%) | Total loss (ha) | Fire-driven loss (ha)) | Total loss (ha) | Fire-driven loss (ha) |
| 2002 | 12,646 | | 4,961 | | 86,179 | 21,890 | 184,140 | 29,713 |
| 2003 | 7,043 | | 2,552 | | 53,578 | 18,693 | 104,882 | 23,135 |
| 2004 | 32,885 | | 12,587 | | 158,904 | 62,232 | 288,634 | 71,538 |
| 2005 | 33,795 | | 9,170 | | 140,345 | 42,260 | 244,178 | 56,281 |
| 2006 | 54,313 | | 12,023 | | 224,249 | 85,081 | 320,690 | 88,869 |
| 2007 | 34,218 | | 6,905 | | 203,990 | 61,875 | 303,782 | 67,606 |
| 2008 | 27,376 | | 876 | | 252,538 | 31,337 | 355,449 | 47,793 |
| 2009 | 29,229 | (1) | 2,543 | (0) | 335,246 | 62,356 | 446,635 | 79,842 |
| 2010 | 6,267 | (8) | 306 | (0) | 120,598 | 14,330 | 228,111 | 28,634 |
| 2011 | 7,105 | (23) | 308 | (42) | 240,864 | 22,776 | 316,644 | 34,771 |
| 2012 | 9,163 | (25) | 495 | (25) | 334,453 | 45,787 | 512,886 | 80,585 |
| 2013 | 6,628 | (50) | 480 | (82) | 176,080 | 21,815 | 245,738 | 32,635 |
| 2014 | 7,264 | (82) | 774 | (96) | 195,885 | 31,298 | 302,012 | 48,848 |

**Fig. 1.**

Table B3: Total and fire-driven forest loss for oil palm expansion in certified plantations in Malaysia and Papua New Guinea during 2002-2014. All areas are given in hectares (ha).

| Year | Malaysia | | | | Papua New Guinea | | | |
|------|----------|---|---|---|------------------|---|---|---|
| | Total loss (ha) | Post-Certification loss (%) | Fire-driven loss (ha) | Post-Certification loss (%) | Total loss (ha) | Post-Certification loss (%) | Fire-driven loss (ha) | Post-Certification loss (%) |
| 2002 | 14,870 | | 912 | | 3,959 | | 1,244 | |
| 2003 | 6,563 | | 791 | | 1,645 | | 301 | |
| 2004 | 13,522 | | 1,912 | | 3,279 | | 721 | |
| 2005 | 6,410 | | 506 | | 1,242 | | 252 | |
| 2006 | 12,312 | | 465 | | 2,893 | | 718 | |
| 2007 | 12,045 | | 15 | | 2,099 | | 479 | |
| 2008 | 7,381 | (2) | 91 | (0) | 1,188 | (34) | 116 | (7) |
| 2009 | 15,467 | (8) | 69 | (0) | 938 | (71) | 3 | (0) |
| 2010 | 10,378 | (19) | 155 | (8) | 716 | (85) | 14 | (96) |
| 2011 | 8,222 | (35) | 120 | (65) | 1,065 | (85) | 4 | (98) |
| 2012 | 7,432 | (48) | 235 | (63) | 1,235 | (79) | 3 | (77) |
| 2013 | 3,261 | (50) | 85 | (78) | 756 | (100) | 0 | (100) |
| 2014 | 4,096 | (82) | 114 | (81) | 477 | (100) | 3 | (100) |

**Fig. 2.**

Table B4: Total MODIS fire detections for certified plantations, including post-certification fire detections.

| Year | Indonesia | | Malaysia | | Papua New Guinea | |
|---|---|---|---|---|---|---|
| | Total fire detections | Post-Certification fire detections (%) | Total fire detections | Post-Certification fire detections (%) | Total fire detections | Post-Certification fire detections (%) |
| 2001 | 169 | | 124 | | 37 | |
| 2002 | 1782 | | 87 | | 130 | |
| 2003 | 716 | | 71 | | 64 | |
| 2004 | 1821 | | 87 | | 130 | |
| 2005 | 1008 | | 128 | | 39 | |
| 2006 | 2712 | | 17 | | 83 | |
| 2007 | 197 | | 12 | | 61 | |
| 2008 | 87 | | 9 | (0) | 43 | (7) |
| 2009 | 483 | (0) | 22 | (0) | 31 | (0) |
| 2010 | 72 | (8) | 18 | (28) | 44 | (95) |
| 2011 | 196 | (29) | 12 | (50) | 18 | (67) |
| 2012 | 191 | (39) | 21 | (33) | 44 | (84) |
| 2013 | 128 | (55) | 11 | (55) | 54 | (100) |
| 2014 | 361 | (73) | 35 | (69) | 52 | (100) |
| 2015 | 656 | (100) | 26 | (100) | 136 | (100) |

**Fig. 3.**

[Figure]

Figure R1: Forest (green) and fire-driven (orange) forest loss within the buffer (5km) areas of certified and non-certified oil palm plantation boundaries. Solid black lines indicate residual forest cover as a percentage of the buffer area adjacent to certified and non-certified plantations.

---

## Author Comment (AC2) · 29 Mar 2017

General Comments: -I would like to see more information on how the area for the buffer analysis was selected. Why were the buffer areas around certified and non-certified plantations combined together? Or would it have made more sense to consider the plantation boundaries vs. plantations+buffers, while also keeping certified and non-certified separate? I'm not sure if you might expect differences in fire activity between buffers around each type of plantation.

We appreciate the Reviewer's suggestion to provide additional information regarding the buffer areas and our analysis of deforestation and fire activity adjacent to oil palm plantations. Reviewer #1 also asked for clarification of the buffer analysis, and raised

several similar questions regarding the potential for differences in buffers for certified and non-certified plantations.

The characteristics of buffer areas around certified and non-certified plantation boundaries were similar (see Figure R1, below), including the patterns of remaining forested area, forest loss, and fire-driven forest loss. In addition, oil palm plantations in Southeast Asia are frequently adjacent to other oil palm plantations (Figure 1), meaning that it is difficult to attribute buffer activities to only certified or non-certified neighbors. As a result, we analyzed fire activity and forest loss for a single set of buffer areas surrounding certified and non-certified plantations.

In the revised manuscript, we would clarify the characteristics of buffer landscapes in Section 2.1, including the fact that nearly 12% of the area within the 5km buffer was mapped as planted oil palm in 2010 (Gunarso et al., 2013; Carlson et al., 2013). Thus, the buffer region may reflect differences in management, in addition to differences in land use and land cover, based on the abundance of planted palm oil outside of large plantations.
* * *
Please also see specific comments below on this topic. -Could there be differences in characteristics besides certification that are influencing the results? It's not clear to me as written if the authors considered other potential variables such as the level of access to plantations, size, whether part of the concession was previously developed, differences in specific provinces, etc. This might also help to address the statistical significance of the results.

We agree that certification is only one of the factors that may account for observed differences in forest loss and fire activity across certified and noncertified plantations. A large literature suggests that when it comes to certification, the producers with the lowest cost of entrance (e.g., the best environmental performance, large producers with sufficient capital) are typically those who become certified (e.g., Garrett et al., 2016).

By not controlling for these factors, we cannot attribute observed lower fire rates to certification. However, our study does not attempt to discern the cause of observed results. Many consumers of palm oil are looking for a commodity with certain attributes (zero-fire, zero-deforestation). Our work informs these conversations because it suggests that RSPO certification is a good signal for such embodied characteristics. Additional studies that control for attributes such as plantation age, size, isolation, and governance are expected to provide further insights regarding the direct influence of certification on environmental outcomes. In our revised manuscript, we will clarify the goal of our study (to measure attributes of certified and not certified palm oil, rather than attributing causality to RSPO certification).

\*\*\*\*\*\*\*\*\*\*\*\*\*\*\*\*\*\*\*\*\*\*\*\*\*\*\*\*\*\*\*\*\*\*\*\*\*\*\*\*\*\*\*\*\*\*\*\*

-Can the authors clarify in the text when they are discussing fires within a year of deforestation (fire-driven deforestation) vs. fires for plantation management/escaped fires? Sometimes it's not clear to me which fire type is being discussed and the description in the methods section does not make this aspect clear.

Our study specifically identifies fires that are spatially and temporally coincident with forest loss, and we describe these fires as contributing to fire-driven deforestation. These fires are distinct from burning for plantation management (e.g., during oil palm replanting), accidental fires either man-made or due to lightening, or fires that occur in non-forest areas. In a revised manuscript, we would revise any wording that might be ambiguous in the description of the fire results.

\*\*\*\*\*\*\*\*\*\*\*\*\*\*\*\*\*\*\*\*\*\*\*\*\*\*\*\*\*\*\*\*\*\*\*\*\*\*\*\*\*\*\*\*\*\*\*\*

Specific Comments:

-Pg. 2, Line 24: What about the % certified within Southeast Asia?

In a revised manuscript, we would clarify that most certified plantations are within Southeast Asia: "By 2016, the RSPO had certified 2.83 Mha of oil palm that produced

10.8 million tons of palm oil, or approximately 17% of global palm oil production, with >90% of certified areas in Southeast Asia (RSPO, 2016)."

-Pg. 3, Line 31: Do you have the date of certification for each plantation or is it only known to have occurred between 2009-2015?

In our original manuscript, the analysis considered the time series of annual fire activity and deforestation for certified and non-certified plantations, based on the extent of certified plantations as of April, 2015. This ever/never treatment of certification did not specifically consider the date of certification for each plantation. The RSPO Principles & Criteria prohibit deforestation of primary or HCV forest after Nov. 2005 and all fire activity, in accordance with laws in Indonesia, PNG, and Malaysia. As a result, our analysis captures the full range of company commitments to sustainable palm oil production covered by the Principles & Criteria, rather than only the actions following the receipt of the RSPO certificate.

However, the date of certification is known for each plantation in our database. In a revised manuscript, we propose to include estimates of the total forest loss, fire-driven forest loss, and total fire activity that occurs on the subset of certified plantations that have already received their RSPO certificate (See tables B2, B3, and B4, below). This detailed breakdown provides a more robust basis for evaluating forest loss and fire activity in certified plantations.

-Pg. 4, Line 4: Was each individual plantation owned by a separate company, or was there overlap in ownership?

RSPO member companies typically have more than one oil palm plantation, although member companies may have both certified and non-certified plantations, as not all plantations must be certified upon joining the RSPO. In our revised manuscript, we will clarify this nested structure.

-Pg. 4, Line 10: Can you give more details on how planted oil palm was detected and

if there were any differences between the three studies?

The three data sources for planted oil palm (i.e., Gunarso et al., 2013; Carlson et al., 2013; TW, 2015) identified oil palm using visual interpretation based Landsat and other high-resolution datasets (i.e., Quickbird). Differences in the date of Landsat or other imagery, including cloud cover, may contribute to potential differences in the estimated extent of oil palm. When multiple estimates were available for the same epoch, we used the combined area from all sources as a more conservative estimate of the extent of planted oil palm.

-Pg. 4, Line 29: How was the 5km buffer selected? Were any differences considered between small vs. large plantations?

We selected a single buffer size (5 km) to evaluate the patterns of fire-driven defor-estation, forest loss, and total fire activity adjacent to palm oil plantations. This buffer was calculated for all plantations combined, given that certified and non-certified plan-tations are frequently adjacent to one another. In general, palm oil plantations in this study were large; in Indonesia, the average size certified plantations (10,700 ha) was comparable to that of non-certified plantations (7,300 ha).

-Pg. 5, Line 12: Could there be any effects of having a 5 year time step for the oil palm datasets vs. the annual deforestation datasets?

The extent of planted palm oil was used to exclude forest loss likely associated with replanting of existing palm oil plantations, rather than clearing of remaining forest area to establish new plantations. Given this approach, estimates of annual forest loss outside of mapped oil palm were considered new forest loss (ie, we assumed that it would be unlikely for planted areas to be established and re-cleared within a single 5-year time step).

-Pg. 5, Line 23: Can you clarify if the certification timing was similar for all of these plantations (2009?) or if it varied across the study area? Could some of the plantations

in the certified category have only been certified towards the end of the study period? If the dates are not known, I would appreciate a discussion at some point in the paper on how this could impact results.

Please see answer above. As noted, the dates of certification vary between 2008 and 2015 for individual plantations. The revised tables (B2, B3, B4) now provide a breakdown of forest loss and fire activity associated with plantations that have already received their RSPO certification, because certification itself (rather than intent to certify) may also impact fire and deforestation dynamics since it includes on the ground visits by auditors.

-Pg. 5, Lines 23-24: Can you comment here or in the discussion on why this could be higher? Were these plantations easier to access or were there other factors that lead to higher deforestation pre-certification? Are these results statistically significant?

Higher rates of forest loss prior to a specific cutoff date (i.e., 2006 or 2009) may indicate an effort to strategically clear forests before restrictions associated with certification begin. The question of statistical significance for differences in clearing rates would necessitate a detailed look at individual plantation characteristics, rather than all certified plantations as a group, in order to control for selection bias of certification. In a separate study (Carlson et al., under review), we evaluate rates of forest loss and total fire activity for matched certified and non-certified concessions.

-Pg 6, Line 1: What do you mean by management classes? Certified, non-certified, and buffers?

Yes, this reference was to certified plantations, non-certified plantations, and buffer areas. In a revised manuscript, we would change this terminology to clarify that this result applied to all three categories of land management.

-Pg. 6, Line 1: Can you mark el nino years on the figure for reference? Any differences depending on the strength of the el nino?

In a revised manuscript, we would mark the El Niño years in Figure 2, as suggested. As the reviewer points out, the strength and duration of El Niño events are somewhat variable (see Figure R2, below). Such differences do influence the total fire activity in different El Niño years (e.g., van der Werf et al., 2008, Field et al., 2016). However, the goal of this work was to compare fire activity across land management classes (certified plantations, non-certified plantations, and buffer areas) in each year, rather than the absolute amount of fire, since oil palm plantations account for a small proportion of total burning in Southeast Asia during El Niño events (e.g., certified plantations in Indonesia account for only 0.5% of all MODIS fire detections in 2015).

-Pg. 6, Line 6: Were the number of dry years consistent between the two periods of comparison?

This particular reference (Pg 6, Line 6) refers to deforestation rates, not fire activity. As a result, we would not expect the number of dry years to influence observed rates of forest loss.

-Pg. 6, Line 15: Again, I'm wondering if you know about differences in certification timing among the three areas?

Yes, as described above, the date of certification is known for all certified plantations in this study. The timing of certification differs among plantations. However, all companies that are members of RSPO agree to the Principles & Criteria of certification and commit to eventually certify all of their mills. The P&C specify reductions in deforestation and fire use that predate the receipt of the RSPO certificate. For example, certification dates for plantations in this study span the period between 2008 to 2015, yet companies with certified plantations joined RSPO as early as 2004. In a revised manuscript, we would clarify the sequence of events that predate certification, including the timing of membership as opposed to certification for member-held plantations.

-Pg. 6: Line 28: Can you give a comparison of the strength of these different El Nino events?

As shown in Figure R2, above, the strength of recent El Niño events is somewhat variable, as documented in a recent studies (e.g., Field et al., 2016). A full exploration of the evolution and duration of El Niño events is beyond the scope of this study (see Figure 5 in Field et al. (2016) for an analysis of precipitation, fire density, and other characteristics of previous El Niño events. The analysis in this study compares fire activity across plantation and buffer classes in each year, but does not compare absolute fire activity across El Niño events where it would be necessary to control for the strength of El Niño events and time-varying aspects of plantation management, including certification.

-Pg. 6, Line 35: I'm not sure I understand exactly what you did here. For the annual fire detections, did you address the difference in temporal sampling between the different datasets? What detection differences might you expect between the different sensors and how could this influence comparisons?

Figure 6 provides an indication of the degree of consistency between MODIS and new high-resolution active fire detections from VIIRS and OLI for 2014 and 2015. The accompanying map panels highlight the additional detail available from higher-resolution observationsâ̆Ťkey advances to support routine monitoring of environmental compliance under RSPO or other certification approaches. The goal was not a validation of current algorithmsâ̆Ťthese questions have been addressed in previous research (e.g., Schroeder et al., 2014). Instead, Figure 6 documents how data from new sensors are consistent with the long-term observations from MODIS and also offer new potential for transparency in monitoring environmental compliance under RSPO or other certification efforts.

-Pg. 7, Line 10: I thought that the Cattau study was focused on concessions that were previously cleared or planted, so wouldn't you expect differences between that study vs. fires used for deforestation as examined here? Or are you considering management fires (see general comment #3)? Not sure if I'm missing something here, so a clarification would be appreciated.

Cattau et al. visually inspected a small number of oil palm concessions (n=53) using data in Google Earth and did not identify evidence of additional palm oil expansion during the period of their study (2012-2015). In contrast, we used satellite-based estimates of forest loss and planted oil palm to separate forest loss and fire activity associated with remaining forest areas from fire detections on existing cleared or planted palm. By reporting both fire-driven forest loss and total fire activity, we are able to separate the fire detections associated with expanding production from other fire types, including intentional management or accidental burning. We are therefore able to address somewhat different questions from Cattau et al., based on the larger sample size of plantations across three countries, longer study period (2002-2015), and separation of fire-driven deforestation from other fire types. Interannual variability in fires associated with forest loss and residual fires related to management or accidental burning (see figure 5) specifically investigates the degree to which fire-driven deforestation occurs on certified plantations, non-certified plantations, and surrounding landscapes in comparison with other fire types. Cattau et al. do not address the question of how fire is used during forest conversion, either as a component of the emissions embodied in certified palm oil or as a source of fires on the landscape during drought years.

-Discussion: If you feel it's warranted, could you comment on whether your work relates to the findings by Gaveau et al. (2016) on the timing of deforestation for oil palm plantations? Gaveau, D. L. A. et al. Rapid conversions and avoided deforestation: examining four decades of industrial plantation expansion in Borneo. Sci. Rep. 1–13 (2016). doi:10.1038/srep32017

Gaveau et al. argue that much of the oil palm expansion in Indonesian Borneo was on previously cleared lands, rather than intact forests. Our study differs from this previous work in several respects. First, we quantified forest loss, fire-driven forest loss, and total fire activity within oil palm plantations. We used existing maps of planted oil palm to isolate changes to remaining forest cover within plantations, and we therefore assume that all forest conversion is for palm oil expansion. In contrast, Gaveau et al. visually

interpreted satellite data to identify planted oil palm for different epochs, similar to data products in our study (e.g., Gunarso et al., 2013; Carlson et al., 2013; TW, 2015). We do not attempt to identify the year of planting relative to the year of forest loss. Second, our study only examines certified plantations in Malaysia, not non-certified plantations. In our study, a higher proportion of new planted palm came from forest in Indonesia (59%) than Malaysia (20%) between 2001-2010. Some differences may be expected in our results based on the extent of plantation areas. In a revised manuscript, we would comment on the difference between our results and the findings from Gaveau et al., while clarifying that different results may reflect the difference in geographic domains and plantation datasets between studies.
* * *
Technical Comments:

-Pg. 3, Line 25: Should it be section 2.1? (Also the rest of the subheadings in this section)

We have changed the section numbers accordingly.

-Pg. 5, Line 1: The VIIRS definition just repeats the first part of the sentence?

We have removed the repeated definition sentence.

-Pg, 5: Line 14: Can you add a supplementary figure show the distribution of peat-lands? We only have the subsets from Figure 1.

In a revised manuscript, we would include the full peatland map as a supplemental figure (see figure R3 below).

-Pg. 5, Line 22: Missing %.

We have included %.

-Pg. 6, Line 29: What were the peak burning months?

August, September, and October. We have added the months in the main text, based on the analysis presented in Figure A3.

-Figure 1: Is it possible to color code the zoomed in subsets by certified vs. non certified? Perhaps with some shading of the peatlands instead? This might make the figure too busy but it would be nice to see the spatial details.

We are unable to provide the information on the location of certified plantations.
* * *
References:

[revised manuscript text omitted]

Peatlands

**Fig. 6.** Figure R3: Extent of peatlands in Indonesia and Malaysia (Wahyunto et al., 2003; 2004;2006 and WI, 2016).

---

## Author Comment (AC3) · 29 Mar 2017

The authors compare fire activity and deforestation between RSPO-certified and non-certified oil palm plantations in Southeast Asia, arguing that RSPO certification has led to reduced fire activity during dry years. This is a well-written paper and the overall result is important. My only significant concern is the assumption that dry conditions during the big fire years were the same with respect to the locations of certified and non-certified plantations.

We appreciate the Reviewer's recognition of the importance of this study. As the Reviewer suggests, the spatial distribution and duration of precipitation anomalies during El Niño events is somewhat variable. In our study, certified and non-certified plan-

tations can only be compared for Indonesia. As indicated in Figure 1, certified and non-certified plantations in Indonesia are clustered in similar locations: 73% of certified plantations were directly adjacent to one or more non-certified plantations, and 89% of certified plantations were within 10 km of a non-certified plantation. Given this clustering, and the spatial resolution of precipitation estimates from the TRMM satellite (0.25 degree resolution), we assume that El Niño events are likely to influence certified and non-certified plantations in a similar fashion. Precipitation distributions vary for recent El Niño events (see Figure R2, below). However, the main emphasis of this study was to compare among certified, non-certified, and buffer areas during the same year. In a revised manuscript, we would further emphasize the relative comparison among certified plantations, non-certified plantations, and buffer areas in a given year.
* * *
Comments P1L21: should this be 'did not stop altogether'?

Yes, we have changed to "did not stop altogether"
* * *
P3L30: I didn't understand the '(ever)' and '(never)' wrt certified and non-certified

In our study, "ever" refers to oil palm plantations that ever got certified after the cutoff year (i.e., year 2009). Whereas, the "never" refers to non-certified plantations—those plantations that were not certified between 2009 and 2015. However, many of the non-certified plantations are in the process of certification and may get certified in future years.
* * *
P4L20: The end of this sentence implies that Southeast Asia has little rainfall seasonality, which I don't think you mean to say.

This sentence refers to the persistent cloud cover in tropical regions without regular dry

seasons, typically considered months with <100 mm rainfall. In a revised manuscript, we would clarify this statement as "regions with persistent cloud cover" to avoid confusion between rainfall seasonality and drier conditions when lower cloud cover facilitates satellite remote sensing.
* * *
P6L28: How are you excluding the possibility that the certified plantations just weren't as dry in 2015 compared to, say, 2006? Figures A3 and Figure 5 clearly show a drop in fire activity over the analysis period over the certified plantations, but from Figure 1, these plantations are not evenly distributed across Sumatra and Kalimantan. It's possible that these regions, for example south-central Kalimantan, were just wetter in 2015 than previous years, given that regional rainfall can vary across El Niño years. Or perhaps they were drier, in which case your argument about RSPO effects is strengthened. Either way, regional rainfall needs to be looked at or mentioned as a possible factor.

We agree with the reviewer that the spatial distribution of plantations is a potential source of variability that was not directly addressed in our original manuscript. In addition to the analysis of proximity, as described above, it is possible to characterize the time series of fire activity for the closest certified and non-certified plantations. In Figure R4, below, we show the time series of MODIS fire detections (similar to Figure 5), restricting the analysis to non-certified plantations within 50 km of one or more certified plantations in Indonesia (N=1076 of 1536 non-certified plantations). By limiting the comparison to nearby plantations, we can more confidently assume that these regions experienced similar precipitation patterns (as shown in Figure R2 using 0.25 degree TRMM data). Figure R4 shows a similar pattern of interannual variability as Figure 5, and the relative comparison among certified and non-certified plantations is consistent with Figure 5 for El Niño years, including 2006, 2009, and 2015. We therefore conclude that the relative difference between certified and non-certified plantations in these years is not driven by differences in the spatial distribution of oil palm plantations.

\*\*\*\*\*\*\*\*\*\*\*\*\*\*\*\*\*\*\*\*\*\*\*\*\*\*\*\*\*\*\*\*\*\*\*\*\*\*\*\*\*\*

P7L9: change 'direct' to 'directly'

We have changed to 'directly'

Please also note the supplement to this comment:
http://www.earth-syst-dynam-discuss.net/esd-2017-2/esd-2017-2-AC3-
supplement.pdf

[Figure]

**Fig. 1.** Figure R2: Maps show monthly totals of the precipitation patterns for Indonesia and Malaysia from Tropical Rainfall Measuring Mission (TRMM, 3B43v7) at 0.25° resolution during the dry season in El Niñ

[Figure]

Figure R4.  Time series of MODIS active fire detections, as in
Figure 5, for certified plantations and the subset of non-certified
plantations within 50 km of a certified plantation in Indonesia.

**Supplement:**

**Reviewer 3: Figures**

[Figure]

Figure R2: Maps show monthly totals of the precipitation patterns for Indonesia and Malaysia from Tropical Rainfall Measuring Mission (TRMM, 3B43v7) at 0.25° resolution during the dry season in El Niño years of 2006, 2009, and 2015. The spatial distribution of dryness was severe in El Niño year 2015, but dry conditions extended until November in 2006.

[Figure]

Figure R4. Time series of MODIS active fire detections, as in Figure 5, for certified plantations and the subset of non-certified plantations within 50 km of a certified plantation in Indonesia.

---

## Author Comment (AC4) · 29 Mar 2017

**Reviewer 2: Figures**

[Figure]

-: Certified :-                    -: Non-Certified :-

- Forest
- Fire related loss
- Residual forest

Figure R1: Forest (green) and fire-driven (orange) forest loss within a 5 km buffer surrounding certified and non-certified oil palm plantation boundaries. Solid black lines indicate residual forest cover as a percentage of the buffer area adjacent to certified and non-certified plantations.

[Figure]

Figure R2: Monthly precipitation for Indonesia and Malaysia from the Tropical Rainfall Measuring Mission (TRMM, 3B43v7) during peak fire months for El Niño years (2006, 2009, and 2015). The spatial distribution of precipitation was similar in 2006 and 2015, whereas the region received more precipitation in October during the 2009 El Niño event.

[Figure]

Figure R3: Extent of peatlands in Indonesia and Malaysia (Wahyunto et al., 2003; 2004;2006 and WI, 2016).

**References:**

Carlson, K. M., Curran, L. M., Asner, G. P., Pittman, A. M., Trigg, S. N. & Marion Adeney, J. 2013. Carbon emissions from forest conversion by Kalimantan oil palm plantations. *Nature Clim. Change,* 3**,** 283-287.

Field, R. D., van der Werf, G. R., Fanin, T., Fetzer, E. J., Fuller, R., Jethva, H., Levy, R., Livesey, N. J., Luo, M., Torres, O. & Worden, H. M. 2016. Indonesian fire activity and smoke pollution in 2015 show persistent nonlinear sensitivity to El Niño-induced drought. *Proceedings of the National Academy of Sciences.*

Garrett, R. D., Carlson, K. M., Rueda, X. & Noojipady, P. 2016. Assessing the potential additionality of certification by the Round table on Responsible Soybeans and the Roundtable on Sustainable Palm Oil. *Environmental Research Letters,* 11**,** 045003.

Gunarso, P., Hartoyo, M., Agus, F. & Killeen, T. 2013. Oil palm and land use change in Indonesia, Malaysia and Papua New Guinea. *Reports from the Technical Panels of the 2nd greenhouse gas working Group of the Roundtable on Sustainable Palm Oil (RSPO)***,** 29-64.

Schroeder, W., Oliva, P., Giglio, L. & Csiszar, I. A. 2014. The New VIIRS 375 m active fire detection data product: Algorithm description and initial assessment. *Remote Sensing of Environment,* 143**,** 85-96.

TW. 2015. *Transparent World-Tree Plantations* [Online]. World Resources Institute: Global Forest Watch. Available: http://data.globalforestwatch.org/datasets/baae47df61ed4a73a6f54f00cb4207e0_5 [Accessed Dec, 09 2016].

van der Werf, G. R., Dempewolf, J., Trigg, S. N., Randerson, J. T., Kasibhatla, P. S., Giglio, L., Murdiyarso, D., Peters, W., Morton, D. C., Collatz, G. J., Dolman, A. J. & DeFries, R. S. 2008. Climate regulation of fire emissions and deforestation in equatorial Asia. *Proceedings of the National Academy of Sciences,* 105**,** 20350-20355.

Wahyunto, B. H., Bekti, H. & Widiastuti, F. 2006. *Peta-Peta Sebaran Lahan Gambut, Luas dan Kandungan Karbon di Papua/Maps of Peatland Distribution, Area and Carbon Content in Papua, 2000-2001.* Bogor, Indonesia.

Wahyunto, R. & Subagjo, H. 2003. *Peta Luas Sebaran Lahan Gambut dan Kandungan Karbon di Pulau Sumatera/Maps of Area of Peatland Distribution and Carbon Content in Sumatera, 1990-2002.* Bogor, Indonesia.

Wahyunto, R. S. & Subagjo, H. 2004. *Peta Luas Sebaran Lahan Gambut dan Kandungan Karbon di Pulau Kalimantan/ Maps of Area of Peatland Distribution and Carbon Content in Kalimantan, 2000–2002* Bogor, Indonesia.

WI. 2016. *Malaysia Peat Lands* [Online]. Available: http://gfw2-data.s3.amazonaws.com/country/mys/zip/mys_peat_lands.zip [Accessed].

---

## Author Response (AR2)

Dear Editor,

We appreciate the additional suggestions from Reviewer #2 to clarify key aspects of our methods and analysis. Below, reviewer comments appear in black text, and our responses follow in blue text. Revisions to the manuscript are highlighted using track changes.

1. Page 4: Did you verify a sample of the plantations with higher resolution data or field observations? Or could you include methods and/or error estimates from the three studies of planted oil palm that are used here?

In the revised manuscript, we have clarified that both Landsat and higher-resolution data were used to map planted palm and validate the data products:

We have added a statement on plantation map accuracy (Pg.4, line17), "Maps of planted palm were generated from 30 m Landsat imagery, and validated using higher-resolution satellite imagery (Carlson et al., 2013; TW, 2015; Petersen et al., 2016)."

Map products range in accuracy from 77% (Carlson et al., 2013), based on a sample of 400 points using QuickBird Imagery, to an overall accuracy of 87% for Malaysia based using very high resolution from Digital Globe (TW, 2015; Petersen et al., 2016). A complete description of the study methods and validation results can be found in the reference papers. Given the potential for map differences, we used the combined extent of mapped palm plantations from all sources, as noted in the text.

2. Page 4: The selection of 5km for the buffer region still seems somewhat arbitrary. Were any sensitivity analyses performed with this distance?

The selection of the 5km buffer was based on expert opinion. Although forest loss could be evaluated using a narrower buffer width, the resolution of the active fire production from MODIS (1 km, plus positional uncertainty) necessitates a larger buffer width to capture fire activity associated with lands that border existing plantations. The total area in the buffer was approximately double the extent of palm oil plantations in Indonesia (Table 1). Expanding the buffer to consider a broader landscape would dilute the information content of land use dynamics on neighboring lands. The goal was to evaluate forest loss and fire activity adjacent to existing palm oil plantations, rather than a landscape-scale assessment of land use dynamics across Indonesia based on multiple buffer sizes or a wall-to-wall assessment.

3. Figure A6: This is a nice figure to show differences in precipitation, but is there a way to include this in the analysis with some metric showing any differences (or not) between certified and non-certified plantations? And is this figure ever referred to in the main text?

We appreciate the Reviewer's suggestion to bring material from our initial response letter into the manuscript. In the revised paper, we have clarified that certified and non-certified plantations experienced similar conditions during El Niño years, based on 1) the degree of

adjacency among plantations and 2) the 0.25 degree resolution of long-term satellite precipitation records.

The revised caption (now Figure A5, based on the new text reference, Pg. 7, line 31) now reads:

"Figure A5: Monthly precipitation patterns for Indonesia and Malaysia from Tropical Rainfall Measuring Mission (TRMM) at 0.25° resolution for months with peak fire activity during the 2006, 2009, and 2015 El Niño events.  Certified and non-certified plantations are clustered in similar locations (see Fig. 1); 73% of certified plantations were directly adjacent to one or more non-certified plantations, and 89% of certified plantations were within 10 km of a non-certified plantation.  Given this clustering, and the spatial resolution of precipitation estimates from the TRMM satellite, we assume that precipitation reductions during El Niño events influence certified and non-certified plantations in a similar fashion."

4. Is there any information on whether it's a handful of plantations dominating the fire driven deforestation or if it's more evenly spread across all concessions

In fact, fire-driven deforestation was detected across a large fraction of palm oil plantations in Indonesia.  As shown below (Table B7), more than 1/3 of Indonesian plantations that would later be certified had fire-driven deforestation between 2002-2007.  From 2008 to 2014, the number of plantations with fire-driven deforestation declined, consistent with lower overall forest loss (Figure 2), but deforestation activity was still distributed across 34-50 plantations.  A larger fraction of non-certified plantations in Indonesia had fire-driven deforestation in all years.  In Malaysia, fire-driven deforestation detections were less common (Tables B1, B5, B7), as was total fire activity (Table B6).

Table B4: Number of certified and non-certified plantations with fire-driven deforestation between 2002-2014. Plantations with fire-driven deforestation after receiving RSPO certification are shown in parenthesis beginning in 2009.

| Year | Certified | | | Non-Certified |
| | Indonesia N=154 | Malaysia N=119 | Papua New Guinea N=10 | Indonesia N=1536 |
| --- | --- | --- | --- | --- |
| 2002 | 63 | 20 | 4 | 747 |
| 2003 | 64 | 12 | 5 | 733 |
| 2004 | 82 | 16 | 4 | 913 |
| 2005 | 78 | 18 | 3 | 859 |
| 2006 | 67 | 12 | 5 | 927 |
| 2007 | 66 | 5 | 5 | 902 |
| 2008 | 39 | 8 | 5 | 724 |
| 2009 | 50 (0) | 7 (0) | 3 (0) | 886 |
| 2010 | 35 (1) | 10 (2) | 3 (1) | 738 |
| 2011 | 34 (6) | 12 (5) | 2 (1) | 697 |
| 2012 | 36 (12) | 9 (5) | 2 (1) | 783 |
| 2013 | 39 (17) | 8 (4) | 1 (1) | 692 |
| 2014 | 37 (25) | 8 (6) | 2 (2) | 766 |

We have added a statement to the main text to clarify that fire-driven deforestation was detected on a large proportion of oil palm plantations in Indonesia (Pg 6, Lines 2-4):
"More than 1/3 of Indonesian plantations that would later be certified had fire-driven deforestation between 2002-2007 (Table B4). As total deforestation declined from 2008 to 2014, fire-driven deforestation activity was still distributed across 34-50 plantations (22-32%, Table B4)."

5. Could you more clearly say something in the discussion about not matching the plantations with ancillary factors (accessibility, etc.) when comparing certified and non-certified? Unless I'm missing it, I couldn't find this in the revised text.

We agree with the reviewer that a matched analysis is one potential way to further investigate the influence of certification on land management. We have a separate paper, currently under review, that uses matching to quantify the influence of certification on forest loss for certified plantations in Indonesia. However, it would be premature to reference this additional work at this time.

We have added a statement to the discussion section to more specifically reference this line of analysis (Pg.8, line 30) :

"In a future study, it may be possible to control for differences in remaining forest cover, plantation age, or company management practices using a matched study design."

Carlson, K. M., Curran, L. M., Asner, G. P., Pittman, A. M., Trigg, S. N. & Marion Adeney, J. 2013. Carbon emissions from forest conversion by Kalimantan oil palm plantations. *Nature Clim. Change,* 3**,** 283-287.

Petersen, R., Goldman, E., Harris, N., Sargent, S., Aksenov, D., Manisha, A., Esipova, E., Shevade, V., Loboda, T. & Kuksina, N. 2016. Mapping tree plantations with multispectral imagery: preliminary results for seven tropical countries. *World Resources Institute, Washington, DC*.

TW. 2015. *Transparent World-Tree Plantations* [Online]. World Resources Institute: Global Forest Watch. Available: http://data.globalforestwatch.org/datasets/baae47df61ed4a73a6f54f00cb4207e0_5 [Accessed Dec, 09 2016].